# Spontaneous mutations and the origin and maintenance of quantitative genetic variation

Wen Huang[1,2,3†], Richard F Lyman[1,2,3†], Rachel A Lyman[1,2,4‡], Mary Anna Carbone[1,2,3], Susan T Harbison[1,2§], Michael M Magwire[1,2,3¶], Trudy FC Mackay[1,2,3*]

[1]Program in Genetics, North Carolina State University, Raleigh, United States; [2]W. M. Keck Center for Behavioral Biology, North Carolina State University, Raleigh, United States; [3]Department of Biological Sciences, North Carolina State University, Raleigh, United States; [4]Department of Biology, Knox College, Galesburg, United States

*For correspondence:
trudy_mackay@ncsu.edu

†These authors contributed equally to this work

Present address: ‡Department of Biology, Washington University in St. Louis, St. Louis, United States; §Laboratory of Systems Genetics, National Heart Lung and Blood Institute, Bethesda, United States; ¶Syngenta, Research Triangle Park, United States

Competing interests: The authors declare that no competing interests exist.

**Abstract** Mutation and natural selection shape the genetic variation in natural populations. Here, we directly estimated the spontaneous mutation rate by sequencing new *Drosophila* mutation accumulation lines maintained with minimal natural selection. We inferred strong stabilizing natural selection on quantitative traits because genetic variation among wild-derived inbred lines was much lower than predicted from a neutral model and the mutational effects were much larger than allelic effects of standing polymorphisms. Stabilizing selection could act directly on the traits, or indirectly from pleiotropic effects on fitness. However, our data are not consistent with simple models of mutation-stabilizing selection balance; therefore, further empirical work is needed to assess the balance of evolutionary forces responsible for quantitative genetic variation.

## Introduction

Mutation is the ultimate source of genetic variation. In natural populations of finite size, however, mutations and the genetic variation they introduce are constantly removed by genetic drift and by purifying and stabilizing natural selection. Therefore, to understand how genetic variation is generated and maintained, one must know the rate by which spontaneous mutations occur and their effects on fitness and quantitative traits, and the forms and consequences of natural selection on genetic variation. To answer these fundamental questions, we take advantage of mutation accumulation (MA) lines of *Drosophila melanogaster* derived from a single inbred genome and independently maintained under conditions of minimal natural selection, as well as a panel of wild-derived inbred lines representing a balanced state of mutations, drift and natural selection. The expectation is that MA lines are experiencing severe genetic bottlenecks thus have minimal natural selection. Therefore, unless a mutation is highly deleterious or lethal, its fate is determined primarily by genetic drift. This property of MA lines is distinctly different from natural populations and allows us to derive the expectation of genetic variation under neutrality using the estimated mutational variance among the MA lines. By comparing the expectation under neutrality to the observed level of genetic variation in a natural population, one can infer the form and consequences of natural selection on genetic variation (*Figure 1*).

We use whole genome sequencing to determine the rate and characteristics of spontaneous mutations, and quantitative measurements of gene expression and organismal traits in the MA and wild-derived lines to understand the origin and maintenance of quantitative genetic variation.

**eLife digest** A key challenge in evolutionary biology is to understand how genetic variation – differences in the DNA of individuals in a population – is generated and maintained to create the enormous diversity that exists in nature. Mutations to the DNA introduce new variation, but these are constantly removed from populations by two other evolutionary forces: natural selection and genetic drift.

Natural selection removes harmful genetic mutations that affect an organism's fitness and reproduction, and genetic drift is the random increase in, or loss of, a genetic variant from a population over time. However, disentangling the effects of these evolutionary forces is challenging because the genetic variation we observe is often the final product of a long history of interaction between them.

Huang et al. have now investigated genetic variation by breeding fruit flies in the laboratory. Natural selection was minimized for these flies; genetic drift was therefore the main force that removed variation.

Huang et al. then sequenced the DNA of the flies to estimate the rate at which genetic mutations spontaneously occur. The sequences contained many more "high-impact" mutations (which directly affect how proteins in the fly's cells work) than seen in sequences taken from a natural fly population.

Traits that are produced by the cumulative actions of many genes and the environment are known as quantitative traits. By examining how much variation genetic mutations introduced into the quantitative traits of each generation of the laboratory-grown flies, Huang et al. estimated how much variation should occur in a natural population whose quantitative traits evolved without natural selection. This estimate was much higher than the levels of genetic variation seen in nature, suggesting that natural selection acts to eliminate mutations that significantly affect quantitative traits.

Simple theoretical models cannot explain the relatively high spontaneous mutation rate and low genetic variation seen in the quantitative traits of natural populations. Therefore, further work is now required to understand more about the balance of evolutionary forces that maintain quantitative genetic variation.

Previous MA studies have successfully used similar strategies to identify mutations and estimate mutation rates in yeast (*Lynch et al., 2008*), algae (*Ness et al., 2012*), nematodes (*Denver et al., 2004*), flies (*Haag-Liautard et al., 2007*; *Keightley et al., 2009*; *Schrider et al., 2013*), and plants (*Ossowski et al., 2010*); and to characterize the nature of natural selection on gene expression (*Denver et al., 2005*; *Rifkin et al., 2005*). However, compared to the present study, they were small in scale, did not simultaneously identify mutations at the DNA level and estimate mutational variance for gene expression and organismal quantitative traits, or did not compare mutational variation to standing genetic variation in the same equilibrium population from which the MA lines were derived.

## Results

### Identifying spontaneous mutations

To accumulate spontaneous mutations, we split one sequenced inbred line from the *Drosophila melanogaster* Genetic Reference Panel (DGRP) (*Mackay et al., 2012*) into 25 MA lines, and maintained them in small populations (10 females and 10 males) for many generations (*Figure 1b*). The small population size will minimize natural selection and allow non-lethal mutations to drift to high frequency or fixation and accumulate over time. We sequenced all 25 MA lines at generation 60 and obtained deep sequence data for 23 lines (*Supplementary file 1A*). We detected a total of 1,456 mutations that were either fixed or segregating at high frequency (>0.20) on the euchromatic nuclear genome in the MA lines. The number of mutations per MA line ranged from 35 to 193, with a mean of 63 (*Supplementary file 1B*). To validate mutations detected by Illumina's sequence-by-synthesis chemistry, we sequenced a randomly selected set of 51 mutations in five MA lines at

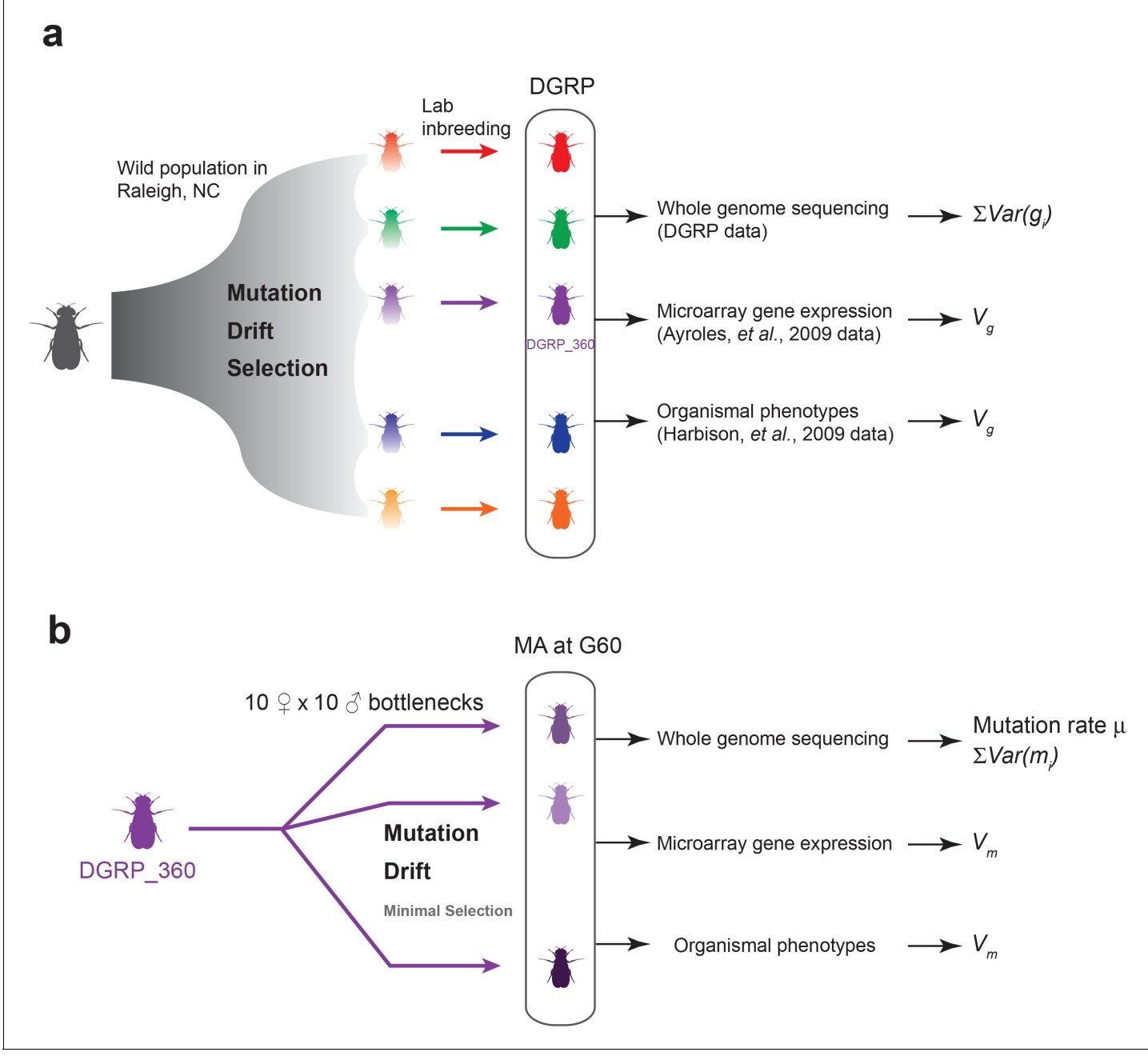

**Figure 1.** Experimental design of the DGRP and mutation accumulation (MA) lines. (a) The DGRP is a collection of inbred strains derived from the wild population in Raleigh, NC. The spectrum of mutations and genetic variation in the DGRP is a reflection of the combined effects of mutations, drift, and selection. In this study, we used the DGRP genotype data (**Huang et al., 2014**) to estimate the parameter $\sum_{i=1}^{G} Var(g_i)$ (see Materials and methods), the microarray expression data collected by (**Ayroles et al., 2009**) and organismal phenotypes (sleep traits from **Harbison et al., 2009**) to estimate genetic variance among the DGRP lines ($V_g$). (b) We derived 25 MA lines from the inbred line DGRP_360. These lines were kept with small population sizes of 10 females and 10 males such that there is minimal natural selection. At generation 60, we sequenced the MA lines to estimate mutation rate ($\mu$) and $\sum_{i=1}^{M} Var(m_i)$, and obtained microarray gene expression and organismal phenotypic data to estimate mutational variance ($V_m$).

generation 130 by capillary sequencing (Sanger). We reasoned that the 70 generations between mutation detection and validation allowed sufficient time for mutations that were segregating at generation 60 to drift to fixation, and hence be more reliably detected by Sanger sequencing. The observed fixation probabilities of putative mutations agreed well with the expected probabilities

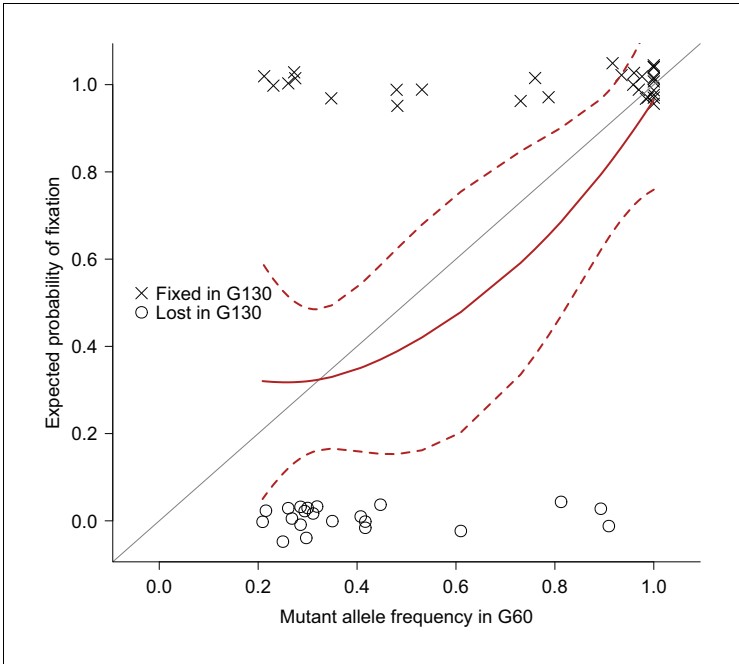

**Figure 2.** Validation of mutations. 51 Mutations detected at G60 (frequency > 0.2) in MA lines 2, 11, 13, 17, or 24 were randomly selected and sequenced by Sanger sequencing at G130 and classified as Fixed (cross) or Lost (circle) by manually inspecting chromatograms. The fixation status is plotted against the initial mutant frequency at G60. A LOESS smooth line was fitted to the data (point estimates = solid line, 95% confidence interval = broken line) to estimate the fixation probability. Expectation of fixation probability (= initial mutant allele frequency) is indicated by the grey diagonal line.

given their initial frequencies at G60 (*Figure 2*), suggesting that both the mutations and their frequency estimates were accurate.

## Genomic properties of spontaneous mutations

Among the 1456 mutations, there were 1203 single base substitutions (SBS), 17 multiple base substitutions (MBS), 141 deletions, 47 insertions, and 48 complex mutational events that involved a combination of base substitutions and indels (*Supplementary file 1B*). The numbers of different types of mutations are proportional to their mutation rates; thus the indel mutation rate was approximately 1/6 that of SBS, and deletions occurred at a frequency three times higher than insertions. Of the 1203 SBS, 595 were transitions (Ti) and 608 were transversions (Tv), corresponding to a Ti/Tv ratio of 1.96 (*Figure 3*). There were twice as many base substitution mutations at G or C sites ($n$ = 828) than at A or T sites ($n$ = 401), which, given the 42.48% genomic GC content, indicated a strong bias (~ 2.80 fold increase) towards mutations at G or C bases. These numbers agreed well with a previous study using the same sequencing strategy of three *D. melanogaster* MA lines, which reported a Ti/ Tv ratio of 1.95 and an approximately two-fold increase in mutations at G/C bases (*Keightley et al., 2009*). We then asked if genomic context affected whether or not a mutation occurred. While there was no appreciable difference in local sequence (20 bp up and downstream) GC content between mutations and randomly sampled sites, indels appeared to occur more often in low complexity regions, where homopolymers and short tandem repeats can often be found and are prone to replication slippage (*Figure 4*).

To characterize the functional effects of spontaneous mutations, we annotated their genomic locations (exonic, intronic or intergenic) and compared the distribution to that of standing variation in molecular polymorphisms in the DGRP, and to the fraction of total genomic sites in each of these categories. Standing variation in the DGRP reflects the demographic history of the natural Raleigh, NC population and is depleted of exonic variants (*Figure 5a*). However, the extent of depletion of exonic spontaneous mutations was much weaker – the spectrum of spontaneous mutations detected

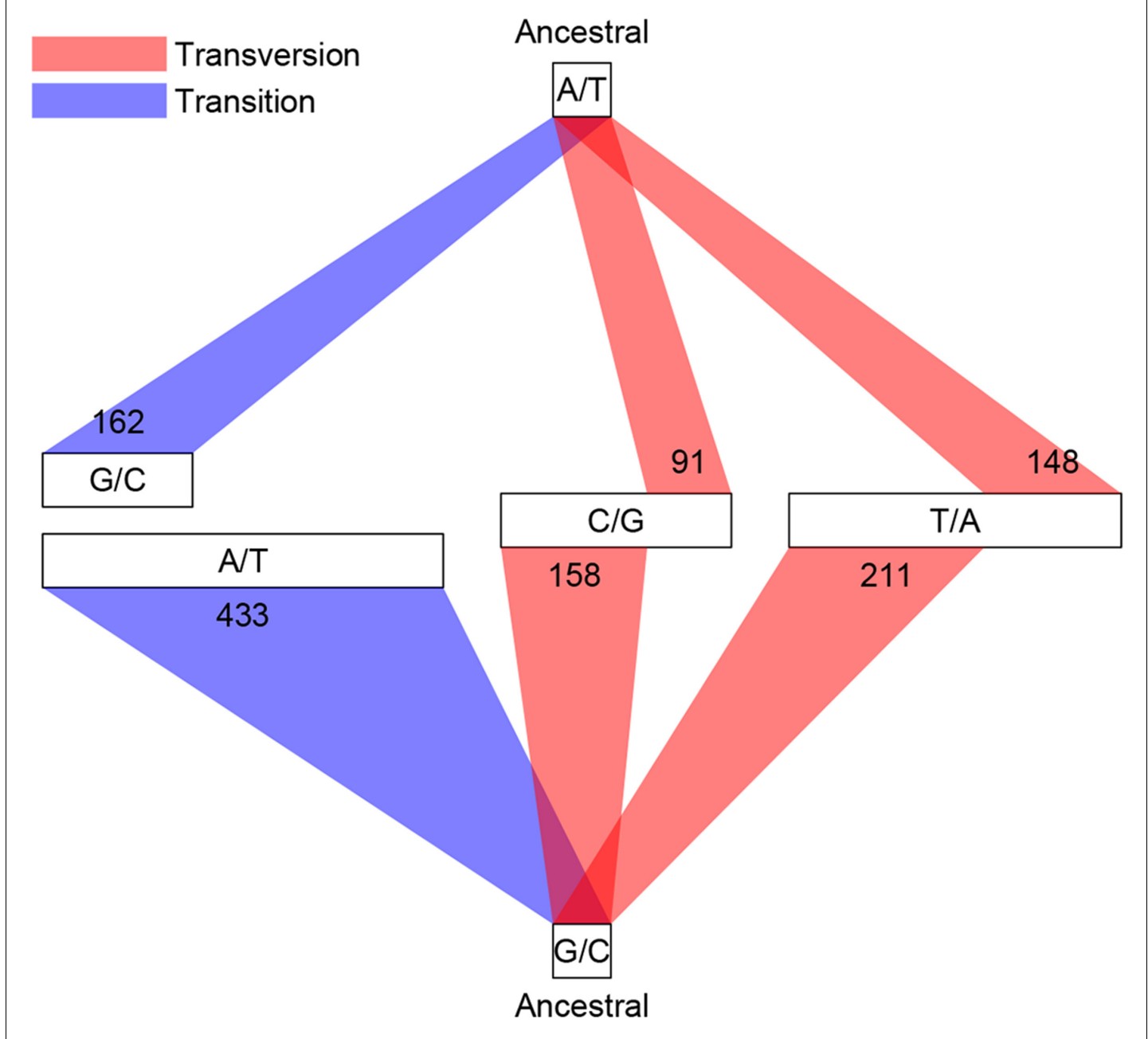

**Figure 3.** Classification of single base substitutions (SBS). Single base substitutions were classified according to their ancestral alleles (top and bottom box) and mutant alleles (middle boxes). The size of each of the middle boxes indicates number of mutations in each class. Blue and red lines indicate transitions and transversions, respectively.

in the MA lines more closely reflected the proportion of exonic, intronic and intergenic bases in the genome (*Figure 5a*). We further classified coding variants and mutations according to their functional impacts on polypeptide sequences. Remarkably, the proportions of frame-shift, stop gained or lost, and nonsynonymous spontaneous mutations were significantly greater than the proportion of segregating polymorphisms in the DGRP for these categories (*Figure 5b*). While the fitness effects of these protein sequence mutations may differ under laboratory and natural environments, these results clearly indicate that deleterious mutations that would otherwise be lost have accumulated in the MA lines. In keeping with the inference that spontaneous mutations accumulated under

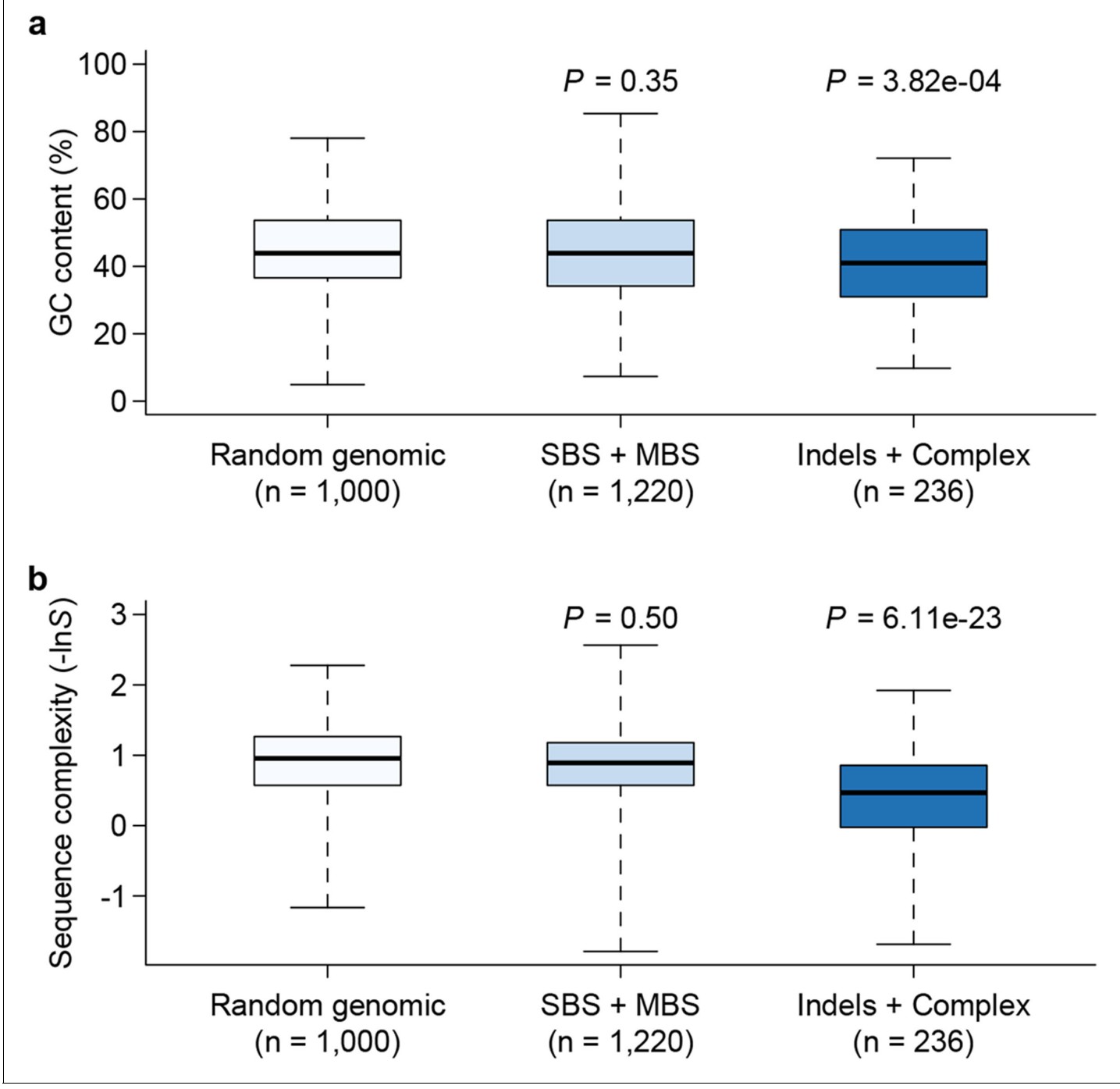

**Figure 4.** Genomic context of mutations. Sequence composition (GC content, **a**) and complexity (**b**) for local sequences (20 bp up and downstream of mutations) are plotted as box plots and compared between different types of mutations. The 'Random genomic' class contains 1000 randomly chosen sites in the genome. Sequence complexity is measured as $-\ln S$, where $S$ is calculated using the algorithm in NCBI's DUST program and measures sequence complexity. $P$ values were computed by Wilcoxon's rank sum tests comparing data in each category to the 'Random genomic' category.

minimal natural selection, the numbers of mutations per gene was primarily a function of gene length (*Figure 6*); and there was no gene ontology (GO) category enrichment for genes harboring new mutations (*Supplementary file 1C*).

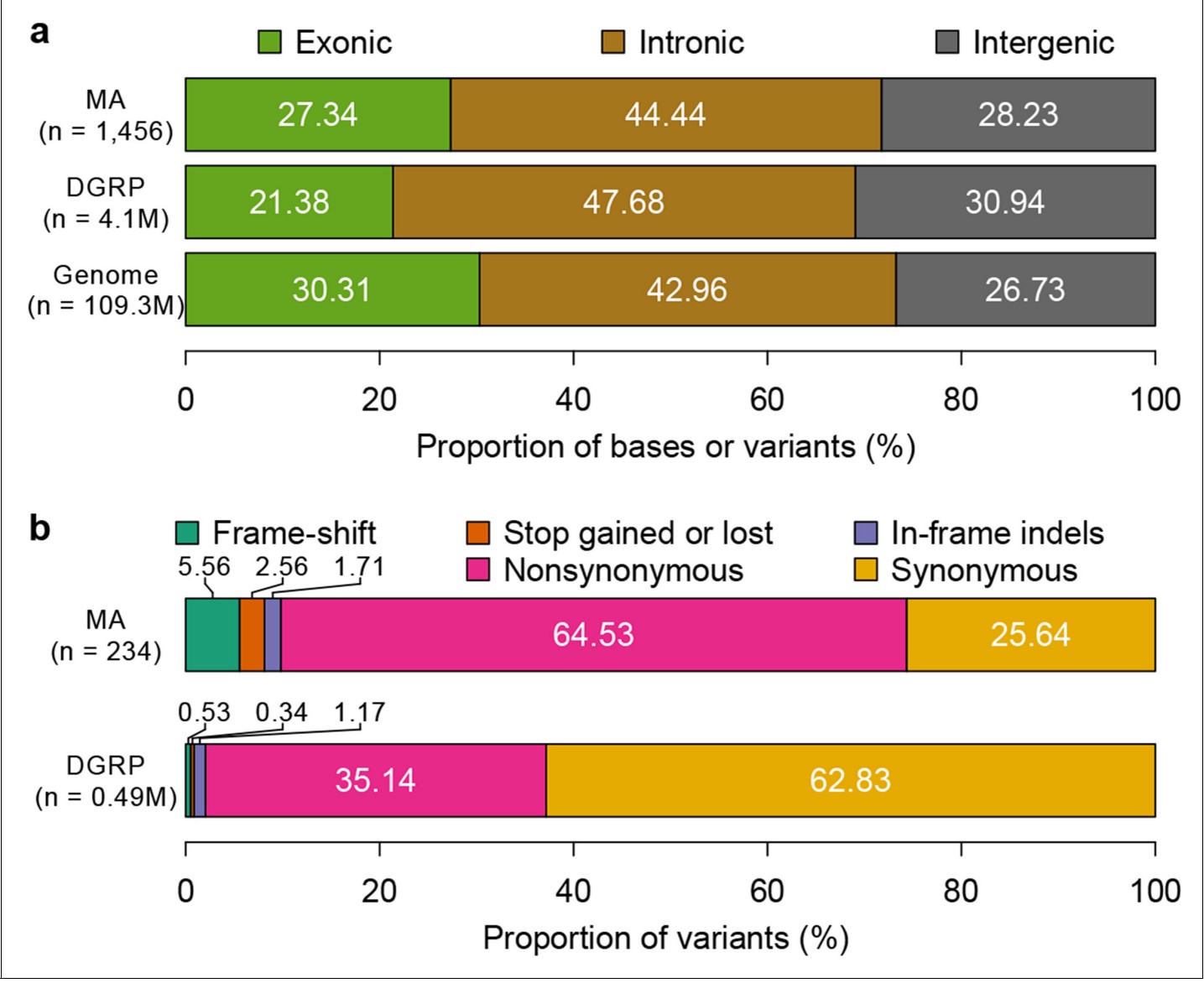

**Figure 5.** Mutation accumulation lines accumulate deleterious mutations. Annotation of genomic bases, standing variation, and mutations in MA lines according to their (a) genomic locations and (b) functional impact on protein sequence.

### Spontaneous mutation rate

To estimate the spontaneous mutation rate, we first inferred the effective population size in the MA lines based on the observed mutation frequency spectrum of presumably neutral mutations. This was achieved by a maximum likelihood procedure where the probability density was obtained by simulation. The parameter $Ne$ was estimated by taking the value that maximizes the likelihood of observing the data given the probability density. Although the census population size of each MA line was $2Ne = 40$ haploid genomes, the effective population size was estimated to be $2Ne = 19$ for the $X$ chromosome and $2Ne = 23$ for autosomes (**Figure 7a,b**). The frequency spectrum of the observed effective population size visually agreed with the expectation (**Figure 7a,b**). At this population size, for selection coefficients $s$ ranging between -0.01 and 0.01 and assuming additivity, the fixation probability $u = \frac{1-e^{-2s}}{1-e^{-4sNe}}$ (**Kimura, 1957**) is between 0.034 and 0.054, which closely centers around the fixation probability for neutral sites. The large difference between census and effective population sizes could be due to the large variance in number of offspring commonly observed in

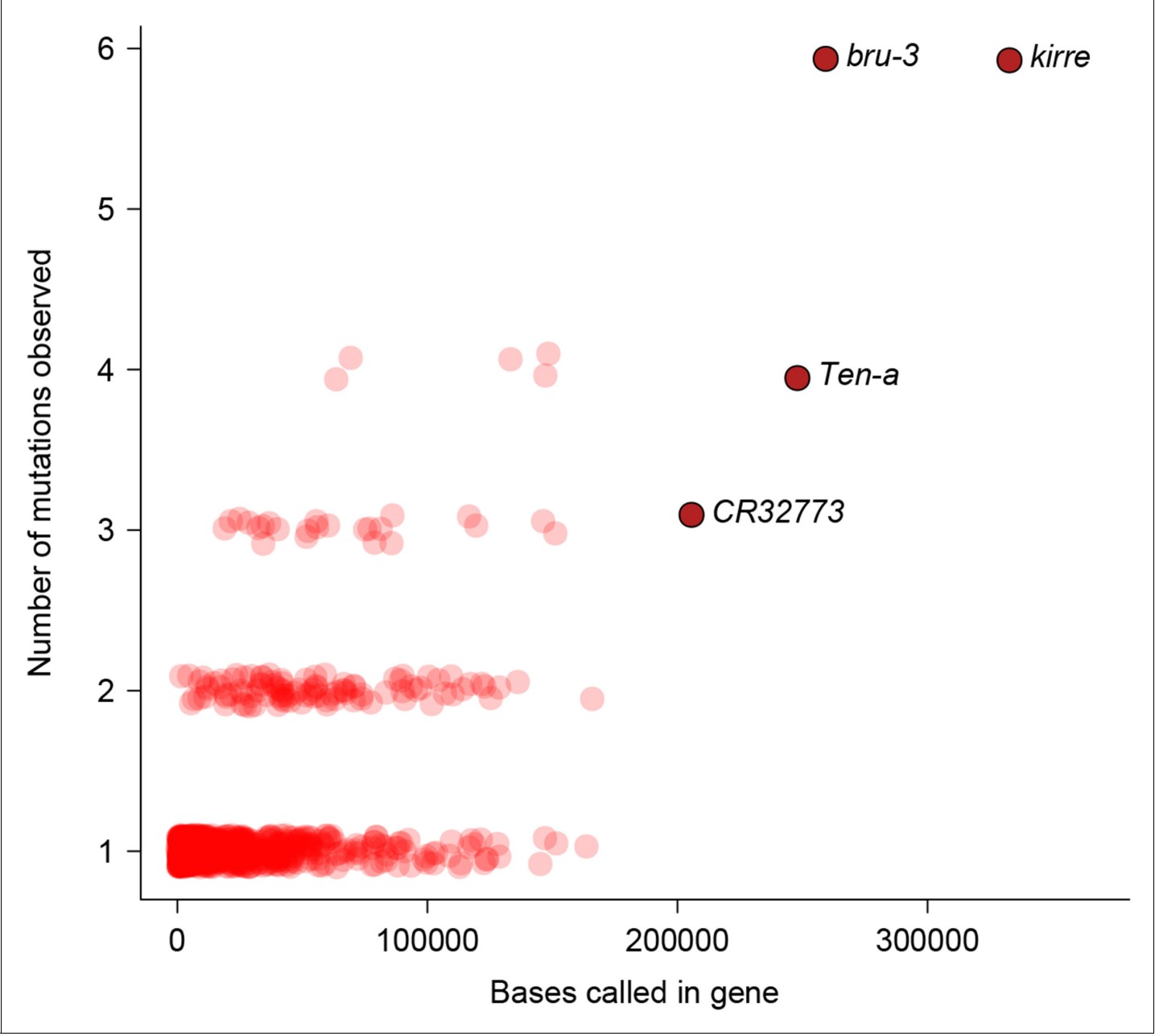

**Figure 6.** Relationship between number of mutations and gene length. Number of mutations detected in MA lines for each gene is plotted against the total number of bases covered for mutations.

flies (***Crow and Morton, 1955***). Although the *X* to autosome *Ne* ratio was slightly higher than the expected 0.75, this difference was not statistically significant (*P* = 0.40). The estimates of *Ne* did not differ whether or not presumably non-neutral mutations (mutations that changed amino acids) were included, further suggesting that drift was the primary force driving mutation frequency dynamics in these MA lines. Given the estimated effective population size, the marginal (integrated over 60 generations) probability of a mutation to attain the frequency cutoff (0.2) was 7.20% on the *X* chromosome and 6.18% on autosomes. We used these estimates of effective population size to estimate the spontaneous mutation rate on the *X* chromosome and autosomes, given the number of observed mutations in each line, taking into account variable sequence coverage in each line.

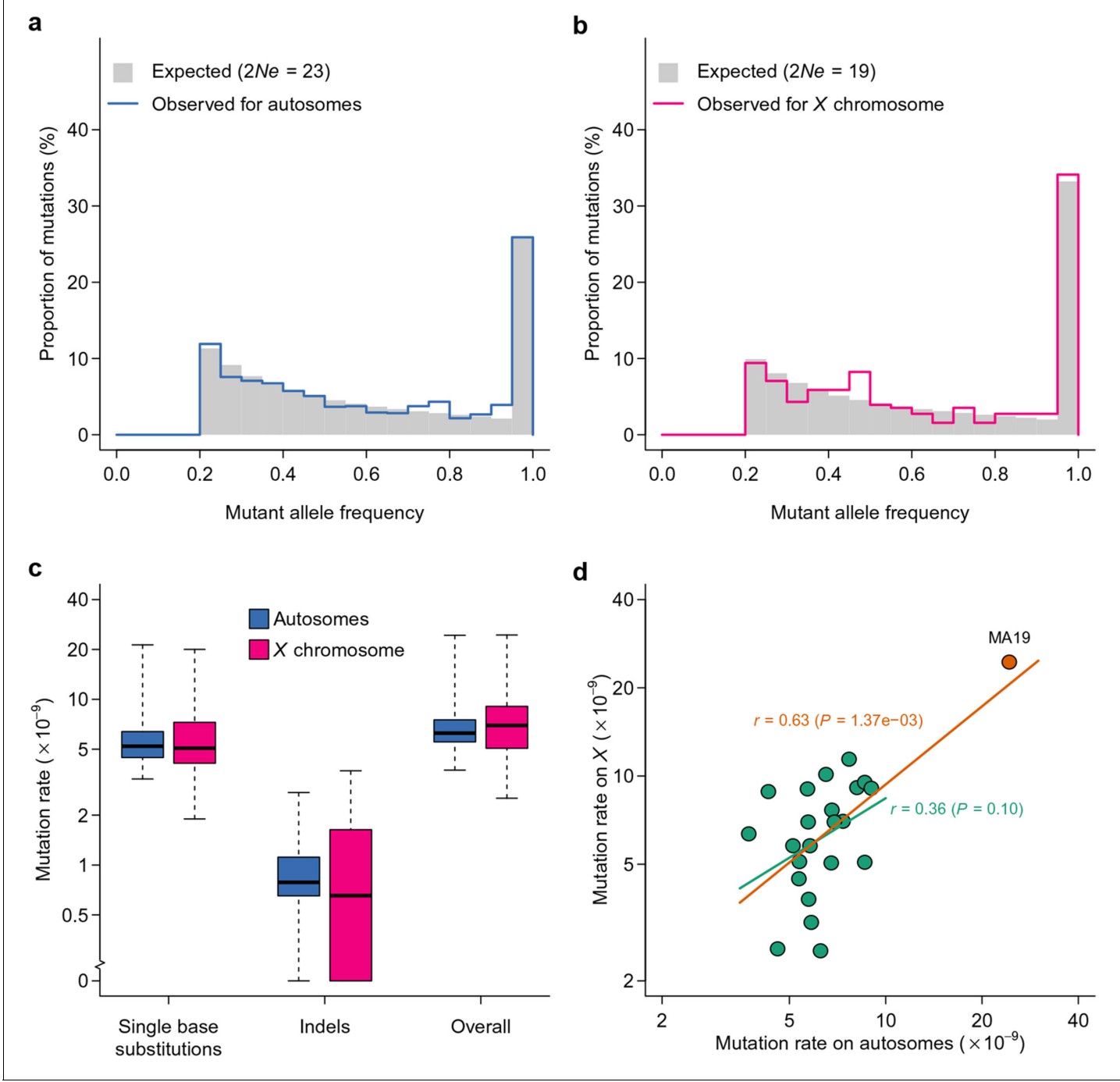

**Figure 7.** Inference of effective population size and mutation rate. Expected and observed distributions of mutant allele frequency on autosomes (**a**) and the *X* chromosome (**b**). The expected distribution was generated based on estimates of effective population size (2*Ne*). Mutation rate estimates of autosomes and *X* chromosomes are compared for different types of mutations (**c**) and for different lines (**d**). In (**d**), the Pearson's correlation coefficient (*r*) is calculated with (orange line) or without (green line) MA19 (orange circle).

The median spontaneous mutation rate was $5.21 \times 10^{-9}$ per base on autosomes and $5.07 \times 10^{-9}$ on the *X* chromosome for single base substitutions and $0.79 \times 10^{-9}$ on autosomes and $0.65 \times 10^{-9}$ on the *X* chromosome for indels. The overall spontaneous mutation rate for all types of mutations combined was $6.25 \times 10^{-9}$ on autosomes and $6.96 \times 10^{-9}$ on the *X* chromosome (*Figure 7c*), similar to recent mutation rate estimates from MA studies using high throughput sequencing

(*Keightley et al., 2009*; *Schrider et al., 2013*). There was substantial variation in spontaneous mutation rates among the MA lines (*Figure 7c*). The mutation rate in MA19 was nearly a magnitude greater than the lines with the smallest mutation rates (MA08 for autosomes and MA15 for the *X* chromosome). Such a large difference cannot be solely explained by variability in $2Ne$ among the MA lines, as varying $2Ne$ from 10 to 40 can only account for a 28% difference in mutation rate (difference in $2Ne * p$, see Materials and methods). The mutation rates on autosomes and the *X* chromosome showed no systematic difference (*Figure 7c*, paired Wilcoxon rank sum test $P = 0.64$) and were positively correlated (*Figure 7d*), further suggesting the random distribution of mutations.

## Rate of introduction of genetic variation by spontaneous mutations

To assess the effects of mutations on quantitative traits, specifically the rate at which mutations introduce genetic variation, we analyzed the genetic variation of genome-wide gene expression profiles, two bristle number traits, and five sleep and activity traits. We partitioned the phenotypic variation observed among the collective sample of individuals from the MA lines into the between-line genetic variation due to mutations ($V_{MA}$) and the within-line variation due to environmental or technical noise ($V_e$). Although any pair of MA lines only differ on average by 126 sites, all organismal phenotypes and the expression of a large fraction of genes in both sexes (1526/7566 = 20.2% of expressed genes in females and 3,872/8,136 = 47.6% in males) accumulated significant between-line variance (*Supplementary file 1D–E*, *Figure 8*). Mutational variance ($V_m$), the amount of genetic variation introduced by mutations in each generation, scaled by environmental variance and expressed as the mutational heritability ($h_m^2 = V_m/V_e$), had a median of 0.55 x $10^{-3}$ for gene expression traits in females and 0.75 x $10^{-3}$ in males (*Figure 8*). $h_m^2$ for organismal traits ranged between 0.30 x $10^{-3}$ and 2.88 x $10^{-3}$, of the same order of magnitude as observed in earlier studies (*Houle et al., 1996*). These values were near or among the upper quartile of that for gene expression traits with the exception of abdominal bristle number in males (*Supplementary file 1D*). The difference in mutational heritabilities between gene expression traits and organismal phenotypes may be due to a larger number of QTLs influencing organismal traits and thus a larger mutational target size.

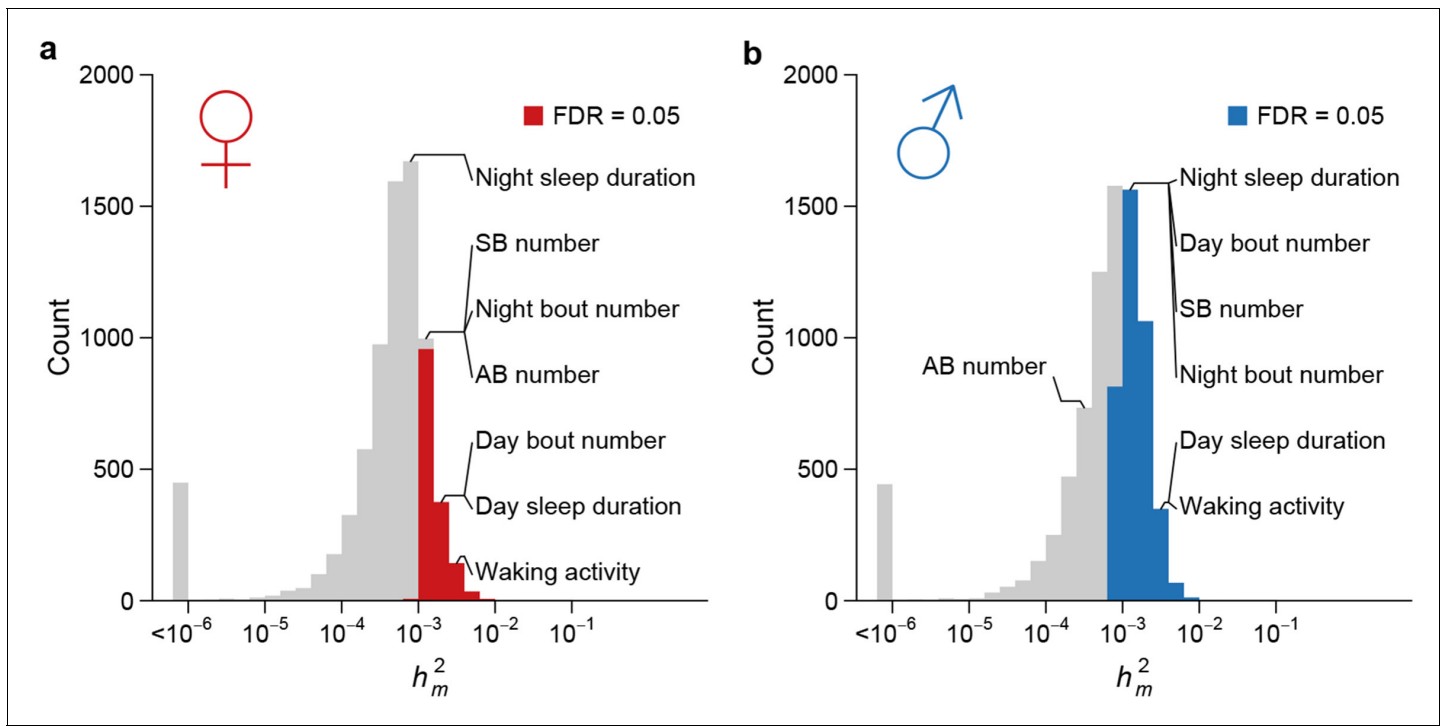

**Figure 8.** High rate of mutational variance. Histogram of mutational heritability ($h_m^2 = V_m/V_e$) for females (**a**) and males (**b**) are plotted on the log scale. The placements of organismal traits in the $h_m^2$ bins are indicated by lines connecting the bars and the trait names. AB = abdominal bristle and SB = sternopleural bristle.

We next assessed whether genes exhibiting mutational variation for gene expression were associated with GO categories. For each GO category, we compared the mutational variance of genes in the category to those not in the category. For significant GO categories, we further polarized the difference in rate of accumulation of mutational variance as faster or slower within the category. Although mutations occurred randomly across the genome and were not associated with GO categories, the rate of accumulation of genetic variation was significantly different for genes within many GO terms than the rest of the transcriptome (*Supplementary file 1F*). For example, in both sexes, mutational variance accumulated faster for genes involved in chitin metabolism, iron binding, and sensory perception of chemical stimulus, but slower for genes involved in protein translation, mRNA splicing, and mitotic spindle organization. Finally, we compared mutational variation in gene expression to plasticity of gene expression across a wide range of environments (*Zhou et al., 2012*). We observed that genes that are more plastic to macro-environmental perturbations accumulated genetic variation at a faster rate (*Figure 9*), suggesting a shared control of gene expression variation by mutations and environmental perturbations (*Landry et al., 2007*).

## Strong stabilizing selection on quantitative trait variation

Under the neutral model of polygenic phenotypic evolution, the among-line variance ($V_g$) in the DGRP inbred lines is $4NV_m$, where $N$ is the effective size of the wild population from which DGRP was derived (*Lynch and Hill, 1986*). We estimated the genome-wide nucleotide diversity in the DGRP ($\pi$) to be $\pi = 4.92 \times 10^{-3}$ in the DGRP (*Huang et al., 2014*), from which we estimated $N$ to be 186,363 ($\frac{\pi}{4\mu}$). For all organismal and the majority of gene expression traits, the magnitude of standing genetic variation in the DGRP was much smaller than that predicted under the assumption of neutrality (*Figure 10*). Therefore, there must be strong stabilizing selection that acts either directly on the traits (direct stabilizing selection) or through pleiotropic fitness effects of new mutations (apparent stabilizing selection), which constrains the accumulation of genetic variation by mutations for gene expression and organismal traits (*Denver et al., 2005*; *Rifkin et al., 2005*). To understand the

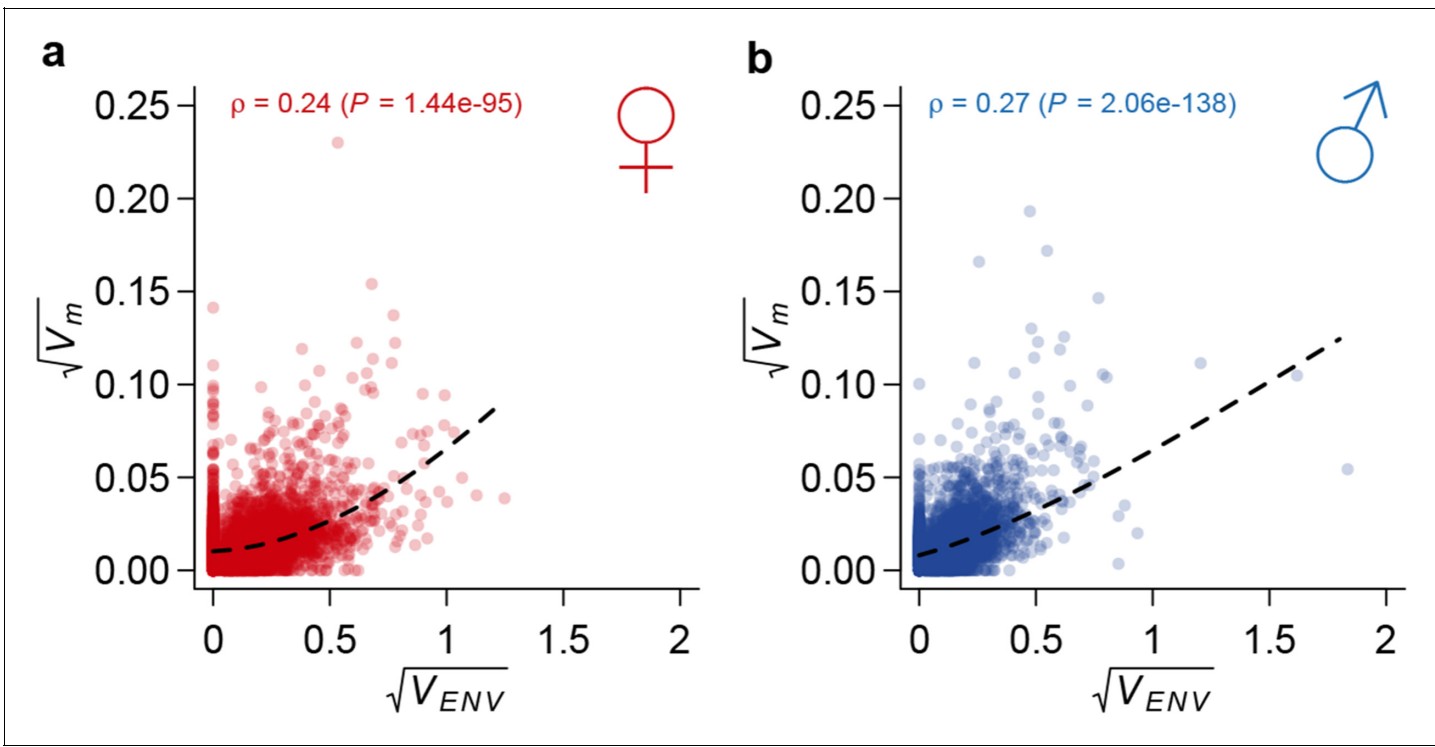

**Figure 9.** Correlation between mutational variance and environmental plasticity. Mutational variance ($\sqrt{V_m}$) is plotted against variance due to environmental plasticity ($\sqrt{V_{ENV}}$) for females (**a**) and males (**b**). The dashed line is a LOESS fit to the data. Spearman's correlation ($\rho$) and the *P* value of a test for its significance are also indicated.

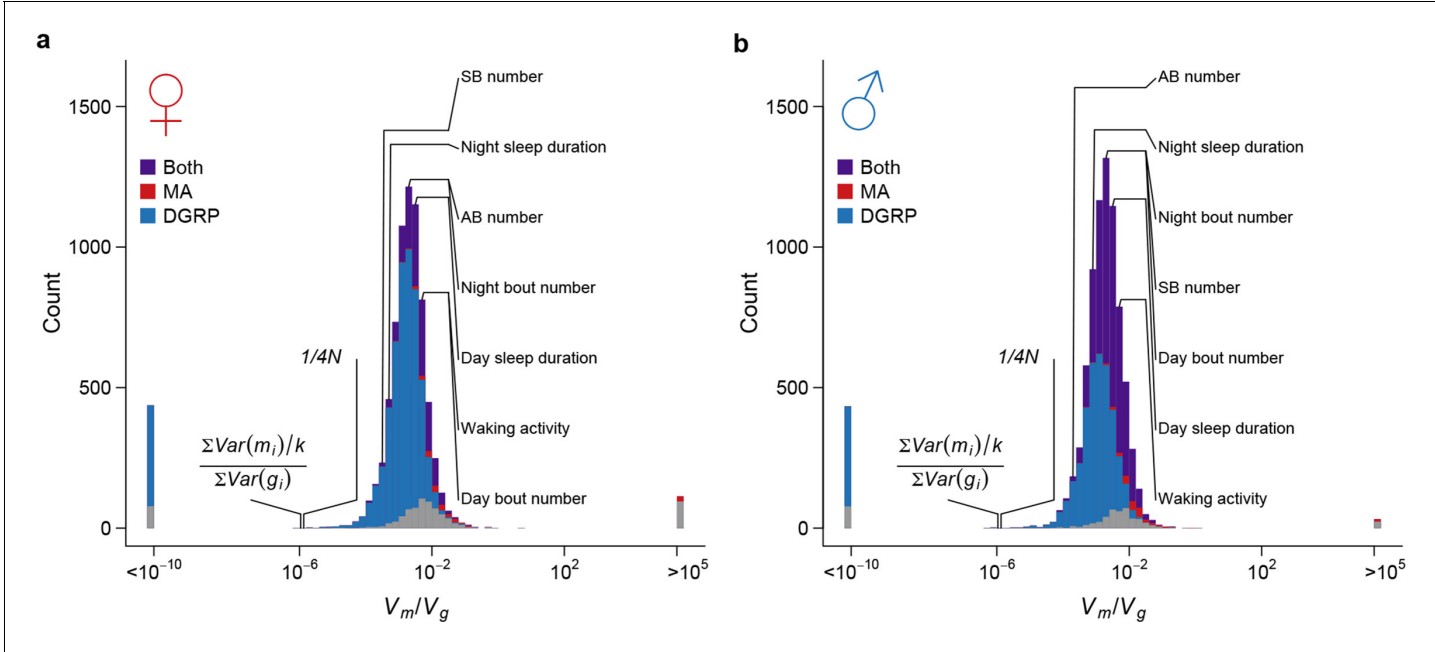

**Figure 10.** Strong apparent stabilizing selection of quantitative trait variation. Distributions of $V_m/V_g$ in females (a) and males (b) are plotted on the log scale. The blue, red, and purple bars indicate genes with significant (FDR = 0.05) among-line variance in DGRP only, MA lines only, and both DGRP and MA lines respectively. Placements of organismal traits are indicated by lines connecting the bars and the trait names. AB = abdominal bristles and SB = sternopleural bristles. Neutral expectations $1/4N$ and $\frac{\sum Var(m_i)/k}{\sum Var(g_i)}$ are also indicated.

consequence of the strong apparent stabilizing selection, we estimated the expectation of allelic effects based on the observed sequence and quantitative trait variation among the MA and DGRP lines. The amount of genetic variation is proportional to sequence variation by a factor of $E(a^2)$ (see Materials and methods), where $a$ is the allelic effect of a mutation on quantitative traits. For the majority of traits, the ratio of mutational to standing genetic variance, $V_m/V_g$, far exceeded the expectation given the observed sequence variation (*Figure 10*); thus the allelic effects of spontaneous mutations, $E(a_m^2)$, were several orders of magnitude larger than that of standing DNA variation, $E(a_g^2)$. This result suggests that the apparent stabilizing selection, directly for fitness and indirectly for traits correlated with fitness, had either eliminated mutations with large effects on quantitative traits or modified their effects. The former appears to be at least partly true given the obvious difference in functional categorization of spontaneous mutations and standing DNA variation (*Figure 5*).

Using $V_m/V_g$ as an indicator of the strength of the apparent stabilizing selection (higher values = stronger selection), we examined the properties of genes associated with variation in $V_m/V_g$ for gene expression. First, genes expressed in both sexes are under slightly stronger selection than those expressed in only one sex, and the strength of selection has a modest but highly significant positive correlation between the two sexes (*Figure 11*). Second, there is stronger selection for genes on the X chromosome than autosomal genes, and this effect is more pronounced in males than females (*Figure 12*). Third, we partitioned the genome with respect to GO categories and assessed the significance of the difference of $V_m/V_g$ for genes associated with each GO category and those not in the category. Many genes in GO categories associated with essential cellular functions related to transcription, translation, cell cycle, and energy metabolism, among others, appeared to be under stronger selection in both sexes (*Supplementary file 1G*, *Figure 13*).

## Discussion

The origin and maintenance of quantitative trait variation are fundamental problems in evolutionary biology and have profound implications in agriculture and medicine, where most economically and

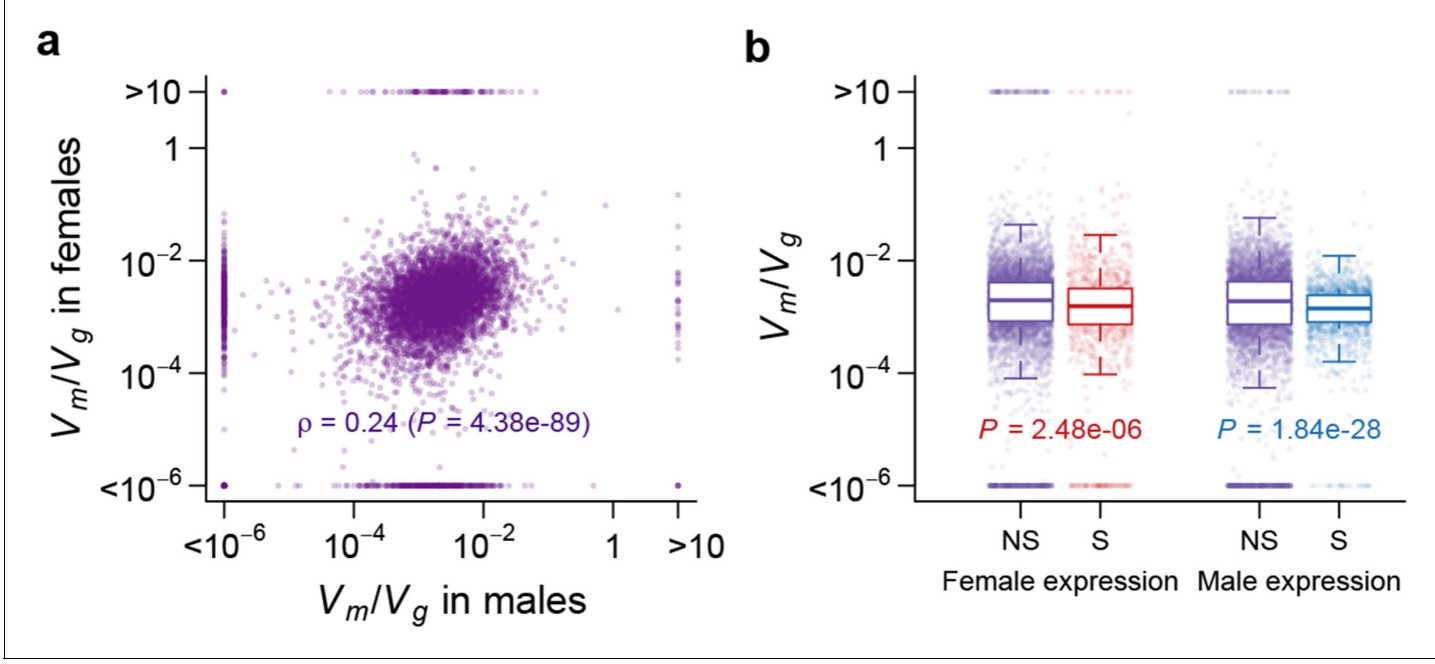

**Figure 11.** Strength of apparent stabilizing selection in females and males. (a) $V_m/V_g$ for gene expression in females are plotted against that in males. (b) Boxplots of $V_m/V_g$ for sex-specific (S) and non-specific (NS) genes. Within each sex, $V_m/V_g$ for sex-specific and non-specific genes are compared using Wilcoxon's rank sum test.

medically relevant traits are quantitative in nature. Mutations are the ultimate source of genetic variation, but they are rare and constantly being removed by natural selection and genetic drift. The purifying property of both selection and drift on mutations makes it especially difficult to study the characteristics of spontaneous mutations in natural populations because it is difficult to attribute a mutational pattern to either one of these evolutionary forces. In this study, we combined the classical mutation accumulation design, which subjects inbred lines of *Drosophila melanogaster* to genetic bottlenecks for many generations, with modern high throughput technologies. This allowed us to largely separate the effects of genetic drift from the combined effects of both selection and drift, a key advantage that is not possible from observations only on natural populations.

Our estimates of mutation rates could be biased either upwards or downwards, depending on the initial fitness of the inbred line. On the one hand, lethal and highly deleterious mutations would not accumulate in these lines. On the other hand, beneficial mutations fix at a higher probability than the neutral probability assumed. Although spontaneous mutations occurred at a relatively low rate of $6.60 \times 10^{-9}$ averaged over autosomes and the X chromosome, they replenished genetic variation at a significant rate for gene expression (sex averaged rate = $0.65 \times 10^{-3} V_e$ per generation) and organismal traits (sex averaged rate = $1.36 \times 10^{-3} V_e$ per generation). Our estimates of $h_m^2$ for gene expression traits were higher than observed in previous studies (*Rifkin et al., 2005*), possibly due to smaller technical variation and thus smaller $V_e$ in this study. Although the MA lines were derived from the same progenitor line and were raised under homogeneous conditions, mutation rates among the MA lines varied significantly. This implies either that non-genetic factors affect mutagenesis and/or that some spontaneous mutations themselves affect mutation rate; the latter is less plausible because such mutations must occur early in many lines to have a pronounced effect. The MA lines accumulated a broad functional spectrum of spontaneous mutations, including a significant fraction of high impact mutations that would otherwise likely be removed by natural selection. The relatively small number of mutations and the large number of genes with significant mutational variance in expression implies pervasive pleiotropic effects by new mutations. Gene expression traits often form co-expression modules (*Ayroles et al., 2009*), and therefore mutations that directly influence expression of a small number of loci can cause secondary *trans* effects at a much larger number of genes. Taking this networked view of gene expression traits, any selection will not act on

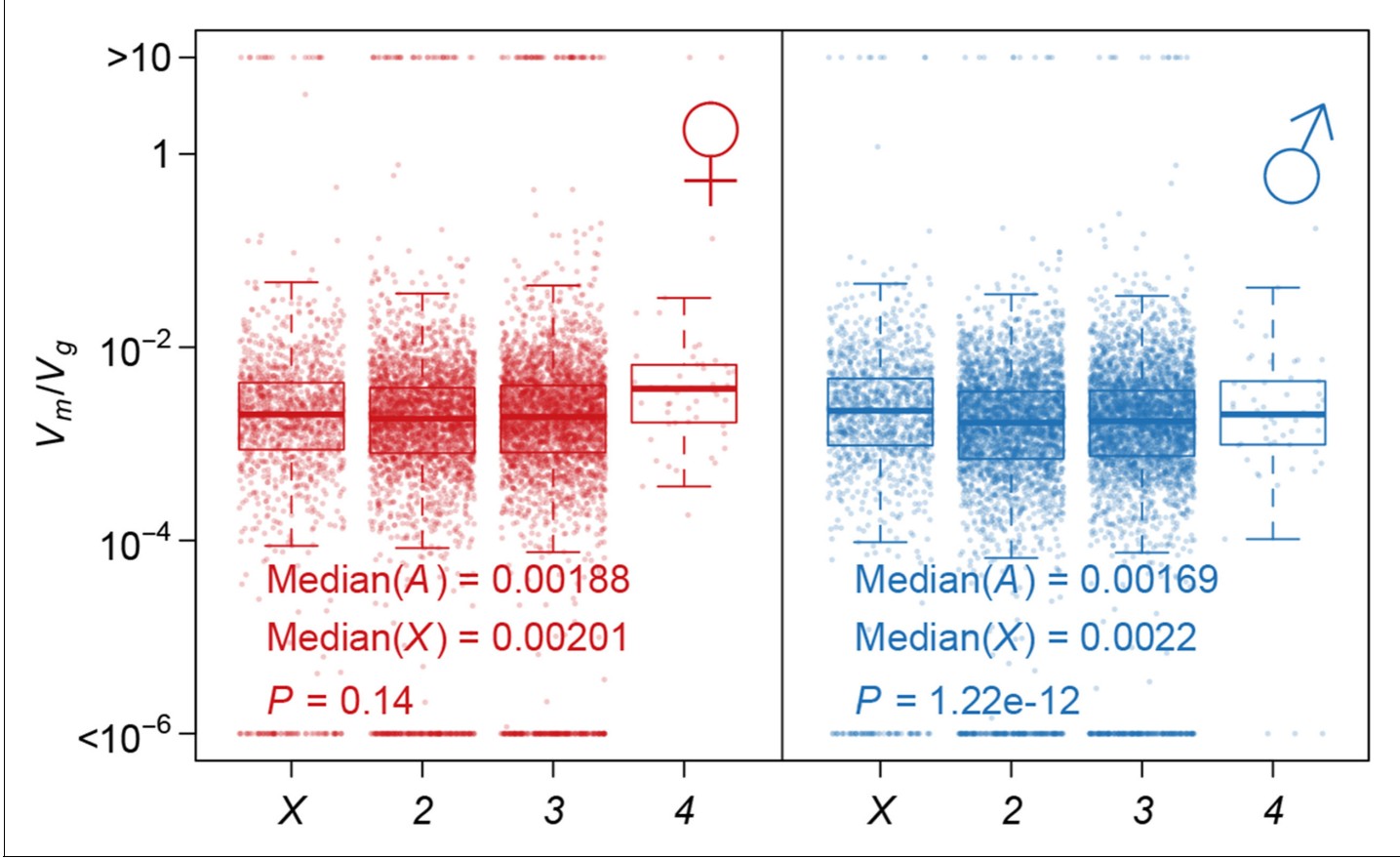

**Figure 12.** Strength of apparent stabilizing selection on autosomes and $X$ chromosome. Boxplots of $V_m/V_g$ are plotted for each chromosome in females and males. Wilcoxon's rank sum test is used to compare $V_m/V_g$ on autosomes ($A$) and on $X$ chromosome ($X$).

individual traits, but rather on the combined effects of all traits. Consistent with this notion, we found stronger selection on genes that would cause *trans* pleiotropic effects such as transcription factors (*Supplementary file 1G*).

Because of the scale of experiment and the large number of MA lines needed to accurately estimate mutational variance, we chose an experimental design that focused on a single inbred strain to derive a relatively large number of MA lines. This design, however, does not allow us to infer the effects of genetic background on the rates of mutation and introduction of genetic variation. There is limited evidence for genuine genetic variation for mutation rate. However, this is primarily due to technical limitations because mutation rate is sensitive to environmental and physiological factors that cannot be easily controlled and it is prohibitive to collect genetic data on mutation rates in natural populations. The effects of genetic background can be studied with a modified design using the DGRP by deriving MA lines from multiple genetically diverse inbred strains and controlling environments. Because of the close proximity of mutation rates in this study and earlier studies of the same species, the conclusions drawn in this study are likely to hold across genetic backgrounds, especially given the magnitude of apparent stabilizing selection, which is unlikely to be attributable to genetic factors unique to the ancestral line.

It is entirely possible that mutation rates may be different in inbred conditions than in natural populations, which has significant theoretical consequences (*Agrawal, 2002*). While we could not formally exclude the possibility, the mutational variance we observed in the MA lines appears to be too large (by a few orders of magnitude) to be solely explained by elevated mutation rates under stressful conditions.

The issue of why genetic variation for quantitative traits segregates at appreciable levels in natural populations despite the tendency of genetic drift and directional and stabilizing selection to

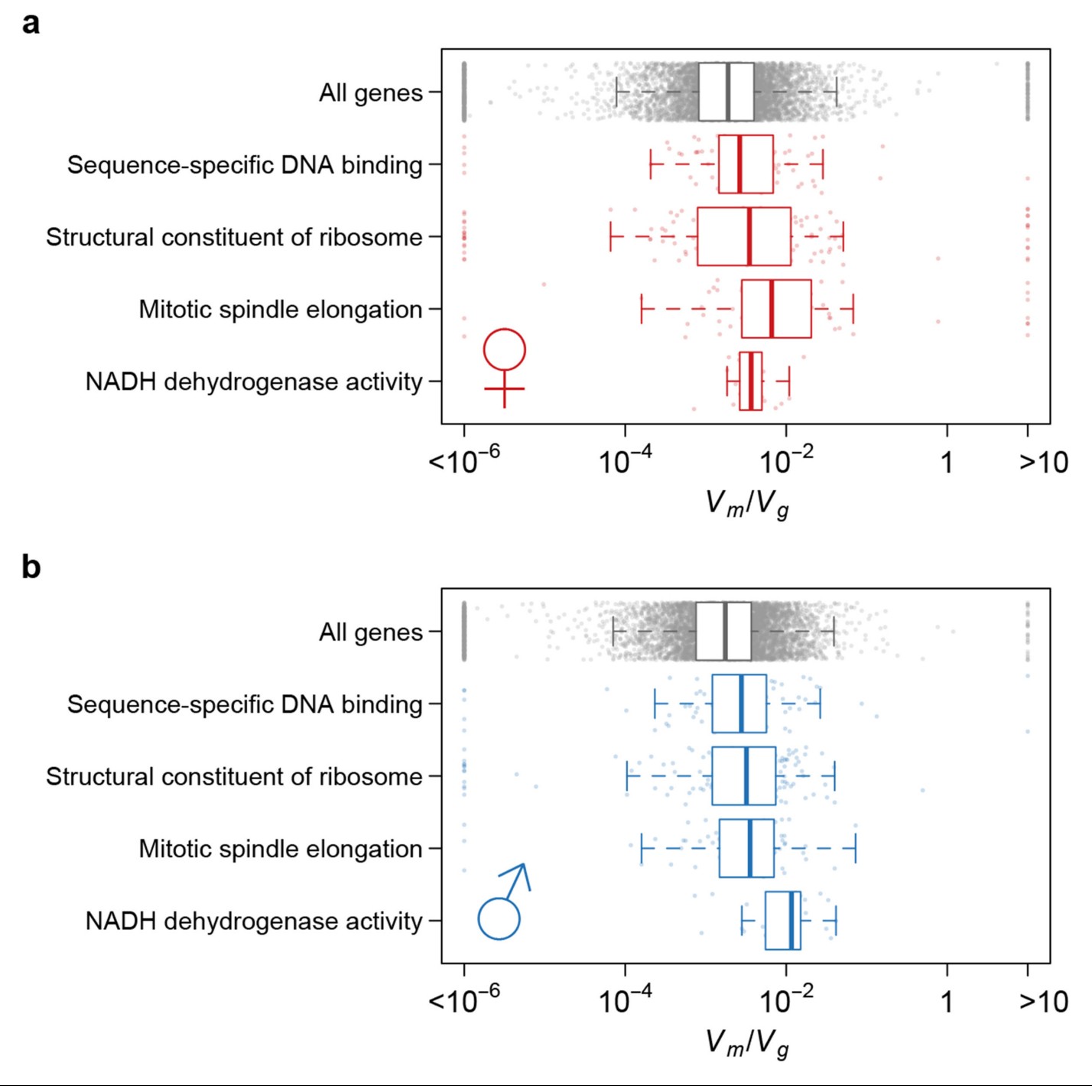

**Figure 13.** Strength of apparent stabilizing selection differs for genes in different functional categories. Boxplots of $V_m/V_g$ for genes in selected Gene Ontology (GO) categories as compared to all genes.

erode it remains an unresolved puzzle. Our data unequivocally reject the neutral mutation – random drift balance model for maintenance of quantitative genetic variation (*Lynch and Hill, 1986*). The observed magnitude of segregating variation is much less than predicted under this model given our estimates of population size in nature and mutational variation. Therefore, we inferred that a form of stabilizing selection on a large fraction of gene expression as well as organismal traits

constrains naturally occurring genetic variation, consistent with earlier studies using a similar analysis in *C. elegans* (*Denver et al., 2005*) and inter-specific variation in *Drosophila* (*Rifkin et al., 2005*).

Theoretical models of maintenance of quantitative genetic variation by mutation – stabilizing selection balance assume either direct stabilizing selection on each trait or stabilizing selection as a deleterious pleiotropic side-effect of new mutations on fitness (*Johnson and Barton, 2005*). The former class of model has different quantitative predictions depending on the relative magnitude of mutation and selection. The house-of-cards approximation holds when mutation rates are low, mutational effects are large, and selection is strong (*Turelli, 1984*); while the Gaussian approximation holds under conditions of weak stabilizing selection, high mutation rates and small mutational effects (*Lande, 1975*). Our observations of low mutation rates and mutational effects that are much larger than standing DNA polymorphisms favor the former parameterization. Under this model, $V_g = 4n\mu V_s$, where $n$ is the number of loci potentially affecting the trait, $\mu$ is the mutation rate, and $V_s$ represents the strength of stabilizing selection (*Turelli, 1984*). Assuming strong direct stabilizing selection (e.g. $V_s = 20V_e$), high heritabilities (e.g. $V_g = V_e$) and our estimated mutation rate ($\mu = 6.60 \times 10^{-9}$), then $n = 1.9 \times 10^6$, which seems implausibly large given the total size of the *D. melanogaster* euchromatic genome of approximately $1.2 \times 10^8$.

Under simple pleiotropic models of apparent stabilizing selection (all mutations are equally deleterious and have a reduction of heterozygous fitness of *s*, and the strength of selection is strong (~$10^{-2}$) against new mutations) the equilibrium genetic variance is $V_g = V_m/s$ and $s = V_m/V_g$. On average, our estimate of $V_m/V_g$ had a median $1.94 \times 10^{-3}$ for organismal traits and $1.82 \times 10^{-3}$ for gene expression traits. Thus, while we detect apparent stabilizing selection on quantitative traits, it is too weak by an order of magnitude for the majority of genes and quantitative traits to be consistent with observed selection against heterozygous effects of new mutations (*Mukai et al., 1972*; *Mackay et al., 1992*). Alternatively, the rate of generation of new mutations is too weak a force to counter strong apparent stabilizing selection, which removes variation faster than it is generated.

Our estimates of $\mu$, $V_m$ and $V_g$ do not alter the conclusion that simple mutation – stabilizing selection models for either direct or apparent stabilizing selection cannot maintain the observed amounts of segregating genetic variation with the observed mutational input and strong selection: the mutational variance is too low and/or the standing genetic variance is too high (*Turelli, 1985*; *Hill and Keightley, 1988*; *Barton and Turelli, 1989*; *Zhang et al., 2002*; *Turelli and Barton, 2004*). It is possible that stabilizing selection in nature is weaker than assumed (*Kingsolver et al., 2001*; *Kingsolver and Diamond, 2011*), and other mechanisms such as balancing selection, fluctuating allelic effects in the face of temporally and spatially varying environments, canalization of mutational effects, and a combination of direct and apparent stabilizing selection all contribute (*Barton, 1990*; *Zhang et al., 2002*; *Turelli and Barton, 2004*; *Zhang and Hill, 2005*). Ultimately, artificially introducing individual mutations or combinations of mutations and assessing their effects, which is now feasible, will be needed to understand the balance between mutations and selection in maintaining segregating genetic variation for quantitative traits.

## Materials and methods

### Mutation accumulation lines

The *D. melanogaster* strain DGRP_360 was generated by 20 generations of strict full-sib mating from an isofemale line derived from the Raleigh, NC USA population (*Mackay et al., 2012*). We divided DGRP_360 into 25 replicate sublines, each maintained at a census population size of 10 virgin females and 10 males per generation. All lines were maintained in shell vials with 10 ml cornmeal-agar-molasses medium at 25°C, 70% humidity and 12 hr/12 hr light/dark cycle.

### DNA extraction, library preparation and sequencing

Genomic DNA was extracted from 100 female flies per MA line at generation 60 using the QIAGEN Genomic-tip 100/G kit (Qiagen, Valencia, CA). The flies were homogenized with a mortar and pestle to a fine white powder using liquid nitrogen and lysed for 2 hr (50°C) with Buffer G2 supplemented with RNAse A (1.5 mg) and Proteinase K (12 mg). The samples were centrifuged at 7000 x g for 30 min at 4°C and the clear lysates applied to a Genomic-tip 100/G that had been equilibrated with Buffer QBT. Once the lysates had passed through the columns, the columns were washed twice with

Buffer QC. Genomic DNA was eluted from the column with Buffer QF, precipitated with 100% iso-propanol and the DNA pellets washed with 70% ethanol. The genomic DNA pellets were re-suspended in 130 μL of nuclease-free water. Purified genomic DNA (1 μg) was fragmented to an average size of 300–400 bp using Covaris shearing (Covaris, Woburn, MA). DNA libraries were prepared from the fragmented DNA using the Illumina TruSeq DNA Sample Preparation Kit (Illumina, San Diego, CA) by following the manufacturer's procedure. The fragmented DNA was subjected to end-repair, adenylation of 3'-ends, ligation of indexed paired-end adapters and PCR-enrichment of the barcoded DNA. The libraries were quantified by qPCR using the KAPA SYBR FAST Master Mix Universal 2X qPCR Master Mix (Kapa Biosystems, Wilmington, MA). The sizes of the PCR-enriched libraries were verified by Bioanalyzer using the high sensitivity DNA chip (Agilent, Santa Clara, CA). We multiplexed and sequenced 5 libraries per lane on the HiSeq 2000 (Illumina). Sequence data are deposited to the NCBI SRA database with the accession number for the BioProject SRP068116.

## Detection of spontaneous mutation in mutation accumulation lines

Sequence reads from the parental line (DGRP_360) and each of the 25 MA lines were aligned to the *Drosophila melanogaster* reference genome (BDGP5) using BWA-MEM with default parameters (*Li, 2013*). Alignments were locally realigned around known indels in the DGRP and around target regions identified across all samples using GATK (*DePristo et al., 2011*). After PCR duplicate removal and base quality recalibration using GATK, overlapping bases from paired end reads were clipped using bamUtil (http://genome.sph.umich.edu/wiki/BamUtil). Finally, alleles (mapping quality $\geq$13, base quality $\geq$13) were piled up using freebayes(*Garrison and Marth, 2012*). Only lines whose median filtered coverage was above 15 were considered for mutation detection. We considered 109,260,235 sites where between 15 and 250 reads were observed in the parental line and in at least 10 MA lines, and no more than 10 possible alleles were observed in all lines combined. A mutation was called if: (1) no read supported the mutant allele in the parental line; (2) the p value for a Fisher's exact test assessing strand bias of alleles was conservatively >0.001; (3) no more than two possible alleles were observed in the mutant line; (4) the mutant allele frequency was greater than 20% in the mutant line; (5) no line other than the mutant line contained the mutant allele at frequency greater than 5%; (6) no more than two other lines contained any reads supporting the mutant allele.

## Validation of mutations by Sanger sequencing

To validate mutations, we sequenced PCR fragments flanking 51 randomly selected mutations in pooled DNA from five MA lines (MA02, MA11, MA13, MA17, and MA24) in G130 using Sanger sequencing. We sequenced DNA in G130 to allow sufficient time for mutations to fix, because Sanger sequencing cannot readily distinguish low frequency polymorphisms from background noise and is subject to bias in PCR amplification of alleles. On average, mutations fix at a probability equal to their initial frequencies after 4$Ne$ generations. Genomic DNA was extracted from 20 females from the MA lines using the Gentra Puregene Tissue Kit (Qiagen). The flies were homogenized with 2 spherical ceramic beads (MP Biomedical) using the TissueLyser (Qiagen) and lysed for 1 hr (56°C) with Cell Lysis Buffer supplemented with RNAse A (1.5 mg) and Proteinase K (12 mg). Proteins were removed from the clear lysates with protein precipitation solution followed by centrifugation. Genomic DNA was precipitated with 100% isopropanol and the DNA pellets washed with 70% ethanol. The genomic DNA pellets were re-suspended in 100 ul of nuclease-free water. Each sample was diluted to 5 ng/ul and subjected to PCR using 95 primer pairs (*Supplementary file 1B*) with the following cycling parameters: 95°C for 2 min followed by 30 cycles of 95°C / 30 s + 56°C / 30 s + 72°C / 30 s followed by a final extension step at 72°C for 4 min. The PCR products were purified using the PureLink Pro 96 PCR Purification Kit (Life Technologies, Carlsbad, CA) and sequenced with each corresponding forward primer using the BigDye Terminator Cycle Sequencing Kit (Life Technologies).

## Estimation of mutation rate in MA lines

To estimate mutation rate in the MA lines, we first inferred the effective population size using a maximum likelihood approach, assuming that synonymous and non-exonic SNPs are neutral and effective population size and mutation rate stays constant over time and among MA lines. For each 2$Ne$ value ranging from 10 to 40, we simulated 1,000,000 MA lines where one mutation occurred independently per line per generation. At G60, frequencies of all 60,000,000 unlinked mutations were

summarized (based on samples of 200 chromosomes in the last generation) to obtain the expected frequency distribution of mutations under a given $2Ne$ at G60, which allowed us to calculate the likelihood of observing the frequency distribution of neutral sites. We calculated the multinomial likelihood $L = \begin{pmatrix} M \\ m_1, \ m_2, \ \ldots, \ m_{16} \end{pmatrix} \prod_{i=1}^{16} P_i^{m_i}$ of the observed mutant frequency distribution, where $i$ indexed one of the 16 equally sized allele frequency bins between 0.2 and 1 ((0.2, 0.25], (0.25, 0.30], ..., (0.95, 1]). $P_i$ is the probability of observing a mutant allele in the $i$th bin given the expected mutant frequency distribution based on simulation, and $m_1, m_2, \cdots, m_{16}$ are the number of mutations in each bin and summed to $M$. We inferred effective population size for autosomes and $X$ chromosomes separately. Because of the small number of mutations in each line, the mutant frequency distribution was summarized across all 23 lines to provide an overall estimate of $2Ne$ across all lines. Mutation rate in each line was then estimated as $\mu = \frac{m}{t*2Ne*p*B}$, where $m$ is the number of mutations, $t = 60$ is the number of generations, $2Ne$ is the effective population size, $p$ is the estimated marginal probability of a mutation attaining 0.2 frequency given $2Ne$, and $B$ is the number of bases considered for mutation calling in that line.

## Acquisition of microarray data

At G60 we assessed whole genome transcript profiles of the 25 MA lines for 3–5 day old males and females, with two biological replicates per sex and line, using Affymetrix Drosophila 2.0 arrays. All samples were harvested between 9–11 am. Whole bodies of 10 flies per sample were homogenized with 1 mL of QIAzol lysis reagent (Qiagen) and two ¼ inch ceramic beads (MP Biomedical) using the TissueLyser (Qiagen) adjusted to a frequency of 15 Hz for 1 min. Total RNA was extracted using the miRNeasy 96 kit (Qiagen) with on-column DNAse I digestion and following the spin technology protocol as outlined in the manufacturer's manual. The RNA was eluted with 45 µL of RNAse-free water. Total RNA was quantified using a NanoDrop 8000 spectrophotometer (Thermo Scientific, Carlsbad, CA). The 100 RNA samples were processed at all stages in a strict randomized design. Fragmented biotin-labeled aRNA were prepared for hybridization to GeneChip Drosophila Genome 2.0 arrays as described in the GeneChip 3' IVT Express Kit user manual (Affymetrix P/N 702646 Rev.5). Briefly, 200 ng of total RNA was reverse-transcribed to synthesize first-strand cDNA. The cDNA was converted to double-stranded DNA and used as a template for in vitro transcription to synthesize biotin-labeled aRNA. The aRNA was purified using magnetic RNA binding beads and quantified using a NanoDrop 8000 spectrophotometer (Thermo Scientific). 12 µg of purified biotin-labeled aRNA were fragmented and hybridized to GeneChip Drosophila Genome 2.0 arrays (Affymetrix). All microarray data have been deposited to ArrayExpress with accession number E-MTAB-4117.

## Preprocessing of microarray data

In addition to the 3' IVT array data that were generated in this study, we downloaded raw microarray expression data from a previous study profiling the transcriptomes of 40 DGRP lines (*Ayroles et al., 2009*) (ArrayExpress E-MEXP-1594), another profiling the transcriptomes of a synthetic outbred population derived from the same 40 DGRP lines under different environmental conditions (*Zhou et al., 2012*) (ArrayExpress E-MTAB-639). Arrays for DGRP_514, MA-21 and MA-23 were not considered because we lacked sequence information for these lines. Probe intensities were corrected for background hybridization using GCRMA (*Wu et al., 2004*) and quantile normalized within each sex. We lifted the probe target alignment to FlyBase annotation (Release 5.57) and retained only those that mapped entirely (can span exon-exon junctions) to constitutive non-overlapping exons and contained no DNA variation among the MA lines or the 40 DGRP lines. The normalized probe intensities were log 2 transformed and estimates of gene expression were obtained using median polish, which adjusted probe effect and removed potential outliers. We did a preliminary normalization and stringently removed arrays that appeared to be contaminated by flies of the opposite sex or contained more than 1% of genes whose expression was more than five standard deviations higher or lower than the mean expression of the same genes across all arrays. A total of 23 out of the 368 arrays were removed before a final normalization was performed on the remaining 345 arrays.

## Analysis of gene expression variance

We analyzed gene expression for each sex within each experiment separately. For each gene, we first transformed the data to be normally distributed by taking the quantiles of a normal distribution with the mean equal to the median expression and the variance equal to the square of the median absolute deviation (by a scaling factor of 1.4824 to account for the expected difference of median absolute deviation and variance, which has no effect on the statistical inference). We defined a gene as expressed if its expression level was above 3.71 in females or above 4.26 in males. These cutoffs were chosen such that by modeling the expression profile as a mixture of two normal distributions, for a gene with expression higher than the cutoff, its probability of belonging to the high expression group was at least twice that of belonging to the low expression group (*Figure 14*). Partition of variance into between-line and within-line variances was performed using the lme4 package in R. It is important to note that the microarray data were collected in three studies such that data sources were completely confounded with the experiments (MA, DGRP, and environmental exposure). Importantly, all estimation of parameters was performed within each experiment such that any between-experiment batch effects that shifted genome-wide gene expression profiles would not affect our results. Batch effects that increased or decreased variances in gene expression, however, may affect the results. Nonetheless, as line or treatment was randomized within each experiment, any such batch effect on variance would be much more likely to affect the within-line variance than the among-line variances that were used for comparison. More importantly, the magnitude of differences in variances, which was also consistent with other studies that assessed the differences within a single batch (*Denver et al., 2005*; *Rifkin et al., 2005*), was too large to be explained by a batch effect.

## Analysis of organismal quantitative trait variance

All organismal traits were scored at generation 61, one generation after the DNA was sequenced and RNA was analyzed. Abdominal bristle number is the sum of the numbers of abdominal chaetae on the two most posterior abdominal sternites, and sternopleural bristle number is the sum of the total number of macrochaetae and microchaetae on the left and right sternopleural plates. Bristle numbers were scored on 10 males and 10 females from each of two replicate vials per line (total $N = 1,000$). Five sleep traits (total sleep duration during the night and day, numbers of sleep bouts during the night and day, and total waking activity) (*Harbison et al., 2009*) were measured on 3–5 day old flies, for 16 virgin males and 16 virgin females per line ($N = 800$). Prior to sleep measurement, all flies were maintained at a constant density of 30 flies per same-sex vial to mitigate the effects of both social exposure and mating on sleep. We recorded seven continuous days of sleep and activity using the Drosophila Activity Monitoring System (Trikinetics, Waltham, MA), which measures the numbers of times each fly crosses an infrared beam. Data from flies that did not survive the

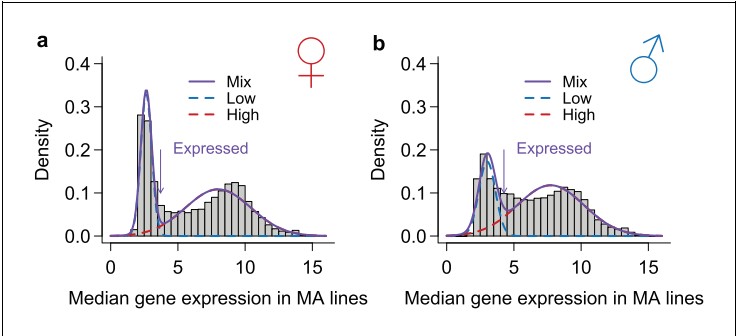

**Figure 14.** Classification of genes as expressed or not expressed in each sex. Within each sex (females in **a**, males in **b**), the distribution of median expression for each gene across the MA lines is subject to a mixture model analysis with two normal components. The mean and variance of each component distribution is estimated using the mixtools package in R. A gene is called expressed if its posterior probability of belonging to the normal distribution with the larger mean is higher than 2/3. The histograms show the observed distribution and the estimated normal distributions and their mixture.

entire recording period were not used in the sleep calculations. Sleep duration was calculated as any period of inactivity lasting at least five minutes. Waking activity was calculated as the number of times the fly crossed the infrared beam divided by the total time awake. We also measured bristle numbers in the 39 DGRP lines for which we had expression data or obtained the sleep phenotypic data for the same lines from a previous study (*Harbison et al., 2009*). The same analysis as the gene expression data was performed to partition variance for the organismal quantitative traits for both MA and DGRP lines, except for bristle numbers, for which an additional between-replicate term was also included.

## Variance in quantitative traits due to mutations, standing variation, and environmental variation

The between-line variance for MA lines $V_{MA}$ under a neutral and polygenic model is approximately, $V_{MA} = kV_m$, where $k = 2\left[t - (2Ne - 5)\left(1 - e^{-\frac{t}{2Ne}}\right)\right]$, $t = 60$ (for gene expression traits) or 61 (for organismal traits) is the number of generations since the lines were established, $Ne$ is the effective population size ($2Ne = 21$) by taking the average of autosomes and $X$ chromosomes and $V_m$ is the mutational variance (*Lynch and Hill, 1986*; *Mackay et al., 1992*). We scaled $V_m$ by the environmental variance $V_e$ to obtain the mutational heritability $h_m^2 = V_m/V_e$. For gene expression traits, the within-line variance is $\frac{V_e}{n} + V_r$, where $n = 10$ is the number of flies per sample and $V_r$ is the technical variation or measurement error of microarrays. $V_r$ is believed to be small thus we multiplied the within-line variance by $n$ to obtain an upwardly biased approximation of $V_e$, which underestimates $h_m^2$. For organismal traits, $V_e$ was assumed to be the within-line variance (sleep traits) or the sum of within-line and between-replicate variances (bristle numbers).

At equilibrium, the among-line variance between a set of inbred lines such as the DGRP ($V_g$) is approximately $4NV_m$, where $N$ is the effective population size (*Lynch and Hill, 1986*) of the wild population from which the DGRP was derived. Therefore $\frac{V_m}{V_g} = \frac{1}{4N}$ when quantitative traits evolve neutrally. We used $\pi = 4N\mu = 4.92 \times 10^{-3}$ in the DGRP to estimate $N$. Because whole genome sequences are available, we can also derive the amount of quantitative trait variance due to sequence divergence, without the assumption of neutrality. The expectation for among-line variance for a trait is $V_g = \sum_{i=1}^{G} E[Var(a_i g_i)] = \sum_{i=1}^{G} E(a_i)Var(g_i) = E\left(a_g^2\right)\sum_{i=1}^{G} Var(g_i)$, where $a_i$ is the allelic effect expressed as a deviation from the ancestral allele at each locus and has an expectation of $E\left(a_g^2\right)$, $g_i$ is the number of copies of the mutant allele in the $i$th line and can take a value in the range of 0 and 2, with numbers between 0 and 2 (twice of the segregating frequency in that line) used to represent lines where the alleles are still segregating, and $Var(g_i)$ is simply the sample variance of $g_i$. This formulation is insensitive to the sign of $a$ and therefore does not require polarization of alleles in the DGRP. For MA lines, mutations below a frequency of 0.2 were randomly drawn from the expected distribution based on the inferred effective population size. The expectation of among-line variance follows the same form: $V_m = \sum_{i=1}^{M} E[Var(a_i m_i)]/k = \sum_{i=1}^{M} E(a_i)Var(m_i)/k = E\left(a_m^2\right)\sum_{i=1}^{M} Var(m_i)/k$, where $a_i$ is the allelic effect of the mutation and has an expectation of $E\left(a_m^2\right)$, $m_i$ measures the number of mutant alleles in the MA line, and $k$ is as defined above. Therefore $\frac{V_m}{V_g} = \frac{E\left(a_m^2\right)}{E\left(a_g^2\right)} \frac{\sum Var(m_i)/k}{\sum Var(g_i)}$, and the ratio of $V_m$ to $V_g$ measures the difference between $E\left(a_m^2\right)$ and $E\left(a_g^2\right)$. If the effect size distributions were equal, $\frac{V_m}{V_g} = \frac{\sum Var(m_i)/k}{\sum Var(g_i)}$. Finally, we estimate variance in gene expression traits due to environmental variation by the among-treatment variance ($V_{ENV}$) in a study of DGRP derived flies subject to 20 diverse and potentially harsh environments (*Zhou et al., 2012*).

## Additional information

### Funding

| Funder | Grant reference number | Author |
|---|---|---|
| National Institute of General Medical Sciences | R01 GM45146 | Trudy FC Mackay |

The funders had no role in study design, data collection and interpretation, or the decision to submit the work for publication.

### Author contributions

WH, RAL, MAC, STH, MMM, Acquisition of data, Analysis and interpretation of data, Drafting or revising the article; RFL, Conception and design, Acquisition of data, Analysis and interpretation of data, Drafting or revising the article; TFCM, Conception and design, Analysis and interpretation of data, Drafting or revising the article

### Author ORCIDs

Trudy FC Mackay, http://orcid.org/0000-0002-2312-7245

## Additional files

### Supplementary files

• Supplementary file 1. Supplemental tables. (A) Summary of DNA sequencing for MA lines. (B) List of detected mutations. (C) GO enrichment analysis of mutations. (D) Variance components of organismal traits among MA and DGRP lines. (E) Variance components of gene expression traits among MA and DGRP lines. (F) Comparison of mutational variance for genes within GO categories to that of genes outside the GO categories. (G) Comparison of apparent stabilizing selection for genes within within GO categories to that of genes outside the GO categories.

### Major datasets

The following datasets were generated:

| Author(s) | Year | Dataset title | Dataset URL | Database, license, and accessibility information |
|---|---|---|---|---|
| Wen Huang, Richard F Lyman, Rachel A Lyman, Mary Anna Carbone, Susan T Harbison, Michael M Magwire, Trudy FC Mackay | 2016 | Mutation accumulation lines generation 60 | http://www.ncbi.nlm.nih.gov/bioproject/PRJNA305361 | Publicly available at NCBI BioProject (accession no: PRJNA305361) |
| Wen Huang, Richard F Lyman, Rachel A Lyman, Mary Anna Carbone, Susan T Harbison, Michael M Magwire, Trudy FC Mackay | 2016 | Gene expression profiles of 25 Drosophila melanogaster mutation accumulation lines | https://www.ebi.ac.uk/arrayexpress/experiments/E-MTAB-4117/ | Publicly available at EBI ArrayExpress (accession no: E-MTAB-4117) |

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
