## [Decision Letter]

Thank you for submitting your article "Spontaneous Mutation and the Origin and Maintenance of Quantitative Genetic Variation" for consideration by *eLife*. Your article has been reviewed by three peer reviewers, and the evaluation has been overseen by a Reviewing Editor and Diethard Tautz as the Senior Editor.

The reviewers have discussed the reviews with one another and the Reviewing Editor has drafted this decision to help you prepare a revised submission.

The following individuals involved in review of your submission have agreed to reveal their identity: Bill Hill (peer reviewer) and Nick Barton (peer reviewer).

As you will see, all three reviewers agreed that the study represented a highly substantial undertaking and that while the results are not surprising, they are an important addition to the literature (as are the data). However, the reviewers also had a number of suggestions that must be addressed for the paper to be published in *eLife*.

1) The substantive concern is that only one MA line is used to estimate the mutation rate and thus that, in the presence of mutation rate variation among lines, the comparison to the DGRP lines is problematic (see Reviewer 2's comments).

This concern needs to be addressed, minimally by making this limitation of the study design explicit in the Discussion, and ideally by providing some quantification of its possible effect.

2) Two reviewers point out that the term "stabilizing selection" is too loosely used.

3) Perhaps most importantly for the dissemination of the work, the paper is not written in a manner accessible to people outside the field. As suggested by Reviewer 3, the expectations need to be more clearly set up, and the methods described in greater detail.

The reviewers also had a number of other suggestions to be taken into account; in particular, it would be helpful to discuss the possibility that MA lines have higher mutation rates than do natural populations. These follow.

Reviewer #1:

This paper does three things:

1) It estimates the spontaneous mutation rate in *Drosophila* from a collection of 25 sixty generation mutation accumulation lines;

2) Measures Phenotypes for several characters (gene expression and a half dozen or so quantitative traits);

3) Measure the SAME set of phenotypes on DGRP lines.

An estimate of the spontaneous mutation rate (at SNPs and small INDELs) is obtained. Furthermore the variation in measured quantitative traits in the MA lines relative to DSPR lines is too large to be consistent with those traits evolving neutrally.

In terms of estimating the mutation rate in a collection of MA lines. This experiment has been carried out in several other systems and even in *D. mel*. (as the authors point out; Keightley et al. 2009 and Schrider et al. 2013) – albeit the oft cited *D. mel* study of Keightley only examines 3 MA lines. Nonetheless, it does not appear that the estimate of mutation rate dramatically changes.

The second conclusion that quantitative traits (QTs) must be under stabilizing selection (because the ratio of natural variation to variation in the MA lines is too low) is not all that surprising. Studies have already concluded as much for gene expression traits by comparing standing variation to divergence (Rifkin et al., 2005 seems to be the go to citation for this). And of course it is not surprising that the quantitative traits are under stabilizing selection. Bristle are sort of a classic QT, that they are under stabilizing selection is discussed in the "textbook" of Falconer and Mackay. It is perhaps a novel observation that sleep and activity traits are under stabilizing selection, but not really surprising (thinking of humans sleep time has a mean of 8 hours and SD of 1 hour, it is easy to imagine sleeping 6 SD's more or less is highly deleterious).

Barring these earlier studies being flawed in some way, this paper doesn't really cover new ground. Overall this paper doesn't really make any novel and/or surprising discoveries, unless I am missing something.

That being said, a really valuable aspect of the paper is that everything is done better than previous studies. And the same MA lines used to measure mutations are used to measure phenotypes, and the same set of phenotypes is measured in the same way in the MA lines and DGRP lines. Furthermore the PI (Mackay) has an incredible track record of quality control, and putting the data out there so anyone can easily download and use it. This work seems like no exception. So what we have is a much better and cleaner dataset that is freely available and useful, and in my mind this is an important contribution.

Reviewer #2:

In this paper from Trudy Mackay's group, a highly inbred line which is one of the DGRP inbreds of *Drosophila* derived as an iso-female line from an outbred population, is used as source from which 25 replicate mutation accumulation (MA) lines were derived and maintained for 60 generations of random mating with 10 parents of each sex. Observations on a number of quantitative traits including bristle number and gene expression and DNA sequences were obtained on all (or nearly all) MA lines. Thus the authors could estimate spontaneous mutation rates for variables ranging from molecular sequence to phenotypes of quantitative traits. The variances and distributions of traits among these lines were compared with those among the DGRP lines and differences used to infer the degree of selection operating in the new lines.

The whole idea seems to me a very nice one, and the scale of this experiment is enormous. The results, whatever they were, are bound to be of general interest and by their analyses and interpretation they add a lot more. I strongly recommend publication, after a relatively small amount of revision. I do, however, have a number of comments, the first being what I think is my main concern with the paper and could be argued is a fatal flaw.

1) A single line (DGRP 360) was used to generate the new MA lines. One significant observation from these lines was that, inter alia, the base mutation rates differed among the new MA lines (Figure 6). This of course implies that comparisons between mutation rates for traits such as bristle number also presumably vary among lines. The evidence for and estimates of e.g. stabilising selection on the phenotypic traits, notably bristle number, were obtained by comparing the variance among the DGRP lines with that predicted from these mutation rates in the MA lines. As there is evidence of between line variance in the MA lines derived from a single DGRP line, there is every reason to expect that there would be differences in the mutation rates among the DGRP lines themselves. Hence the estimates of mutation rate and of stabilising selection effects in comparing the MA and DGRP lines is, presumably, sensitive to choice of DGRP line.

I think the authors need to consider what I think is a serious flaw in the design of the experiment and thus underestimation of the sampling variance and overestimation of significance levels of some of the results. A better design would, in some respects, have been to have taken a subset of MA lines from several (a sample of) DGRP lines to assess and allow for genetic variation in mutation rates among the DGRP lines. There are obviously other benefits in terms of comparisons in sequence etc. by just using one DGRP line, but the problem of lack of replication remains nevertheless. At the very least the authors need to discuss this: the flaw will be apparent to others also.

Reviewer #3:

This is a very substantial study that measures accumulation of mutations in both sequence and quantitative traits in the same mutation accumulation lines, and compares them with standing variation in the original natural population. The analysis is fairly sophisticated (though not well explained), and the estimates are remarkably close to those from previous studies, and estimated by less direct methods.

Overall, this is a large-scale study that addresses an important issue: whether quantitative variation is maintained by a balance between mutation and selection. The results are not surprising, and support the consensus view, but that should not count against this work. On balance, I think that it should be suitable for *eLife*, but it does need some considerable revision – mostly, to explain what was done more clearly to a reader who has not been immersed in these issues.

Generally, the paper is written for readers thoroughly familiar with these kinds of experiment. There is some explanation in the Discussion, but it needs to come much earlier. There needs to be a clearer statement early on about what is expected in mutation accumulation experiments. Highly deleterious mutations will not be seen, but weakly deleterious mutations will accumulate in the MA lines, and will be more frequent than in nature.

The estimates here do not include the fraction of strongly deleterious mutations. In particular, one expects recessive deleterious mutations to potentially rise to moderate frequency in these inbred lines. This would not necessarily show up in Figure 1 if the bulk of mutations are nearly neutral, but could be tested for by asking whether the MA lines can be made homozygous. A comment at least is needed on this issue.

A pervasive problem is the use of the term "stabilising selection" to refer to any kind of purifying selection. It would be clearer to restrict it to selection on a continuous trait towards an intermediate optimum, and it needs t be stead always whether this is selection on the observed or on hidden traits. In any case, it is simpler to see *V_m_/V_g_* as an estimate of net selection against the alleles that contribute trait variance, and avoid speculation about whether this acts via stabilising selection (in the sense defined here).

Results, first paragraph: It would be interesting to compare the frequency of accumulated mutations with the spontaneous mutation rate, measured in other studies that compare parents with offspring – the difference being that the MA lines will not contain highly deleterious mutations.

A serious worry is whether mutation rate is higher, and different in kind, in inbred lines than in the wild population – cf Agrawal's recent work.

Results,second paragraph: "Generally agreed well" is rather vague – one needs a statistical test of whether the rate and spectrum of mutations is the same as in previous work (as SI)

Subsection “Spontaneous mutation rate”: Given the population size, what is the fixation probability as a function of h and s? As it is, the reader has to work this out for herself.

In the same section, there should be a bit more explanation of how the mutation rate is estimated in the main text (a sentence, at least). This is not at all straightforward.

Subsection “Rate of introduction of genetic variation by spontaneous mutations”: What is meant by "global gene expression"? Even after reading the Methods, I could not work this out. It seems bizarre to pool the expression level of all the genes, because, I assume, expression of individual genes will be analysed elsewhere. Still, it is not clear what "global gene expression" is measuring, especially since the expression measurements were made separately in the three experiments and so are not directly comparable.

Second paragraph of the same section: What are the units of measurement here? It is not obvious how to compare variance in gene expression, but the definition of the trait is not even stated in the main text. In the Methods, it emerges that there has been a very obscure normalisation procedure. Are we comparing the variance of absolute values? The variance on a log scale? The variance relative to *V_e_*? This is crucial to how we interpret these variances.

Subsection “Strong stabilizing selection on quantitative trait variation”: Seeing lower standing variance than expected under neutrality does not imply stabilising selection; there could instead be purifying selection against alleles that have pleiotropic effects on the measured traits. The term "stabilising selection" needs to be defined carefully.

In the same section, the comparison of E(am2)
*vs*
E(ag2)is confusing. The implicit assumption here is that stabilising selection acts on the observed traits, which is wildly implausible. Rather, *V_m_/V_g_* gives an estimate of the net selection on each allele. This selection may be due to stabilising selection on unobserved traits, and/or to direct selection against the alleles. (The distinction here depends of course on what "traits" one defines). It is confusing to use the term "stabilising selection" here.

The basic theory is explained in the Discussion but needs to come much earlier to avoid confusion.

Discussion, paragraph four: The estimate of selection from *V_m_/V_g_* is an order of magnitude smaller than estimates of selection against deleterious mutations generally. However, this is consistent with a broad distribution of effects on fitness and on traits: the alleles that contribute to genetic variance will tend to be less deleterious than spontaneous mutations. Also, of course, balancing selection may contribute genetic variance.

Subsection “Estimation of mutation rate in MA lines”: Does the actual frequency distribution of (presumed) neutral sites actually fit the simulation model? Figure 6 suggests yes, but this is not commented on in the text.

It seems rather inefficient to just record the number of mutations at p>0.2 – surely using the actual frequency would be more informative? One would also like to see whether non-synonymous mutations are less likely to reach high frequency, as expected if they are recessive deleterious.

Subsection “Variance in quantitative traits due to mutations, standing variation, and environmental variation”: This formula for VMA makes no sense; it will be negative for large times. There is no explanation of where it comes from.

It is annoying that the figures are all given at the end, but are separated from their captions, and are not numbered.

Figure 7 and Figure 9 are especially inscrutable.

[Editors' note: further revisions were requested prior to acceptance, as described below.]

Thank you for resubmitting your work entitled "Spontaneous Mutation and the Origin and Maintenance of Quantitative Genetic Variation" for further consideration at *eLife*. Your revised article has been favorably evaluated by Diethard Tautz (Senior editor), a Reviewing editor, and three reviewers.

All agree that the manuscript has been improved. There are a couple of remaining points that still need to be addressed (see Reviewer 3's comments), either through revisions to the writing or in a brief note explaining why you do not agree with the concern.

See the specific comments below:

Reviewer #1:

All my concerns were addressed in the response and via the inclusion of the new Figure 1.

Reviewer #2:

This is a revision of a major paper I refereed earlier. I am satisfied with the revision, recognising that the major problem about replication of source lines mutation accumulation is not feasible at this stage, and arguably never was in view of resources needed. The authors respond adequately on this point.

It should now be published.

Reviewer #3:

The revision more or less answers my criticisms, though the changes are minimal. I accept that starting from a single line was a reasonable choice, given the effort involved, and this is now explained better. However, I am still not satisfied on a couple of points:

The revisions have been mostly to better explain the experimental design. However, my original criticism was more that the underlying theory is not set out well at the beginning: this has not been addressed.

A specific problem is with a key interpretation of the results: in paragraph seven of the Discussion. it's stated that the estimated selection is too weak to be consistent with classical estimates of selection against alleles that influence viability. I don't think this necessarily follows. If there is a distribution of effects on traits, then the ratio *V_m_/V_g_* will be smaller for traits that are less closely connected with fitness. With rare alleles, and assuming mutation/selection balance, the selection estimated from *V_m_/V_g_* equals the harmonic mean s, weighted by the squared affect on traits. Thus, of the effect on the trait is not correlated with fitness, we have 1/<1/s>, whereas if the trait is fitness itself, we have <s^2>/<s>, which must be larger. So, the differences in *V_m_/V_g_* between viability and gene expression are surely to be expected?

These issues are discussed in Charlesworth's recent PNAS review, which should be not just cited, but used to frame the interpretation of these data.

A point of terminology: I don't think that "apparent stabilising selection" is used correctly. To me, this refers to the reduction in fitness of extreme phenotypes that is observed even for a neutral trait, when trait variation is due to pleiotropic effects of mutations. However, here "apparent SS" refers to the ratio *V_m_/V_g_* under this pleiotropic model. I think that "purifying" or "pleiotropic" selection would be a better term to use.

---

## [Author Response]

As you will see, all three reviewers agreed that the study represented a highly substantial undertaking and that while the results are not surprising, they are an important addition to the literature (as are the data). However, the reviewers also had a number of suggestions that must be addressed for the paper to be published in eLife.

1) The substantive concern is that only one MA line is used to estimate the mutation rate and thus that, in the presence of mutation rate variation among lines, the comparison to the DGRP lines is problematic (see Reviewer 2's comments).

This concern needs to be addressed, minimally by making this limitation of the study design explicit in the Discussion, and ideally by providing some quantification of its possible effect.

We added discussion to the manuscript about this limitation (please also see below our response to each of the reviewers’ comments). The main consideration at the time of designing the experiment was to balance cost and accuracy. While there could be genetic variation in mutation rates, we were more concerned with obtaining accurate estimates of mutational variance across a large number of traits, which required a relatively large number of MA lines. As it turns out, the strength of stabilizing selection inferred based on our mutational variance estimates was too large to be explained by genetic variation in mutation rates.

2) Two reviewers point out that the term "stabilizing selection" is too loosely used.

We have now revised extensively the entire manuscript to improve clarity in what we mean by stabilizing selection. We distinguish real stabilizing selection and apparent stabilizing selection with more qualifying terms whenever possible. We now refer to most of the observations as “apparent stabilizing selection” unless we specifically refer to different models.

3) Perhaps most importantly for the dissemination of the work, the paper is not written in a manner accessible to people outside the field. As suggested by reviewer 3, the expectations need to be more clearly set up, and the methods described in greater detail.

We have carefully followed all related comments by the reviewers that suggest expanding the description of methods and explanation of results, including adding a schematic diagram to outline the experimental design (a new Figure 1). We believe these revisions throughout the text have greatly improved the manuscript and we appreciate the suggestions from the editor and the reviewers.

The reviewers also had a number of other suggestions to be taken into account; in particular, it would be helpful to discuss the possibility that MA lines have higher mutation rates than do natural populations. These follow.

All other comments have also been carefully considered. Revisions were either made accordingly, or detailed responses to the comments are given below.

Reviewer #1:

This paper does three things:

1) It estimates the spontaneous mutation rate in Drosophila from a collection of 25 sixty generation mutation accumulation lines;

2) Measures Phenotypes for several characters (gene expression and a half dozen or so quantitative traits);

3) Measure the SAME set of phenotypes on DGRP lines.

An estimate of the spontaneous mutation rate (at SNPs and small INDELs) is obtained. Furthermore the variation in measured quantitative traits in the MA lines relative to DSPR lines is too large to be consistent with those traits evolving neutrally.

In terms of estimating the mutation rate in a collection of MA lines. This experiment has been carried out in several other systems and even in D. mel. (as the authors point out; Keightley et al. 2009 and Schrider et al. 2013) – albeit the oft cited D. mel study of Keightley only examines 3 MA lines. Nonetheless, it does not appear that the estimate of mutation rate dramatically changes.

The second conclusion that quantitative traits (QTs) must be under stabilizing selection (because the ratio of natural variation to variation in the MA lines is too low) is not all that surprising. Studies have already concluded as much for gene expression traits by comparing standing variation to divergence (Rifkin et al., 2005 seems to be the go to citation for this). And of course it is not surprising that the quantitative traits are under stabilizing selection. Bristle are sort of a classic QT, that they are under stabilizing selection is discussed in the "textbook" of Falconer and Mackay. It is perhaps a novel observation that sleep and activity traits are under stabilizing selection, but not really surprising (thinking of humans sleep time has a mean of 8 hours and SD of 1 hour, it is easy to imagine sleeping 6 SD's more or less is highly deleterious).

Barring these earlier studies being flawed in some way, this paper doesn't really cover new ground. Overall this paper doesn't really make any novel and/or surprising discoveries, unless I am missing something.

That being said, a really valuable aspect of the paper is that everything is done better than previous studies. And the same MA lines used to measure mutations are used to measure phenotypes, and the same set of phenotypes is measured in the same way in the MA lines and DGRP lines. Furthermore the PI (Mackay) has an incredible track record of quality control, and putting the data out there so anyone can easily download and use it. This work seems like no exception. So what we have is a much better and cleaner dataset that is freely available and useful, and in my mind this is an important contribution.

We thank the reviewer for the enthusiasm for this study. We agree with the reviewer that this study indeed does not make surprising observations. Nonetheless, we believe studies like this that revisit old problems with newer technologies are immensely useful. As the reviewer points out, this study contains the most comprehensive and accurate estimates of several important genetic parameters and a useful large and open data set, which are important for further advances in this field. Please note, these are DGRP lines, not DSPR lines. The latter were derived from only eight inbred strains and do not capture the range of natural variation encompassed by the DGRP.

Reviewer #2:

In this paper from Trudy Mackay's group, a highly inbred line which is one of the DGRP inbreds of Drosophila derived as an iso-female line from an outbred population, is used as source from which 25 replicate mutation accumulation (MA) lines were derived and maintained for 60 generations of random mating with 10 parents of each sex. Observations on a number of quantitative traits including bristle number and gene expression and DNA sequences were obtained on all (or nearly all) MA lines. Thus the authors could estimate spontaneous mutation rates for variables ranging from molecular sequence to phenotypes of quantitative traits. The variances and distributions of traits among these lines were compared with those among the DGRP lines and differences used to infer the degree of selection operating in the new lines.

The whole idea seems to me a very nice one, and the scale of this experiment is enormous. The results, whatever they were, are bound to be of general interest and by their analyses and interpretation they add a lot more. I strongly recommend publication, after a relatively small amount of revision. I do, however, have a number of comments, the first being what I think is my main concern with the paper and could be argued is a fatal flaw.

1) A single line (DGRP 360) was used to generate the new MA lines. One significant observation from these lines was that, inter alia, the base mutation rates differed among the new MA lines (Figure 6). This of course implies that comparisons between mutation rates for traits such as bristle number also presumably vary among lines. The evidence for and estimates of e.g. stabilising selection on the phenotypic traits, notably bristle number, were obtained by comparing the variance among the DGRP lines with that predicted from these mutation rates in the MA lines. As there is evidence of between line variance in the MA lines derived from a single DGRP line, there is every reason to expect that there would be differences in the mutation rates among the DGRP lines themselves. Hence the estimates of mutation rate and of stabilising selection effects in comparing the MA and DGRP lines is, presumably, sensitive to choice of DGRP line.

I think the authors need to consider what I think is a serious flaw in the design of the experiment and thus underestimation of the sampling variance and overestimation of significance levels of some of the results. A better design would, in some respects, have been to have taken a subset of MA lines from several (a sample of) DGRP lines to assess and allow for genetic variation in mutation rates among the DGRP lines. There are obviously other benefits in terms of comparisons in sequence etc. by just using one DGRP line, but the problem of lack of replication remains nevertheless. At the very least the authors need to discuss this: the flaw will be apparent to others also.

We thank the reviewer for raising this important point. We entirely agree with the reviewer that mutation rate and mutational variance can be genetically variable and thus specific to each line. We added discussion in the manuscript to acknowledge this limitation and its implications. We did create MA lines originating from multiple DGRP lines but decided to focus on MA lines derived from one progenitor line for the following reasons. The scale of the work is enormous. When deciding between a feasible design of a few ancestors each with a small number of lines and one ancestor with a large number of lines, we chose the latter, aiming for accurate estimates for one line instead of sub-optimal estimates of several lines. In retrospect, especially given the declining cost of sequencing, we could have analyzed more lines from multiple ancestors. Nevertheless, earlier studies using multiple MA lines derived from different ancestors did not suggest a large genetic variation in mutation rate (Keightley et al., 2009). Moreover, because we need to estimate mutational variance accurately for a large number of traits, we are more concerned with their accuracy than mutation rate *per se*. Perhaps most importantly, the level of the observed stabilizing selection was too large to be explained by genetic variation between lines. While we acknowledge the limitation in the experimental design, what we observed are likely to generalize and indeed are corroborated by earlier studies (also see references in the text).

Reviewer #3:

This is a very substantial study that measures accumulation of mutations in both sequence and quantitative traits in the same mutation accumulation lines, and compares them with standing variation in the original natural population. The analysis is fairly sophisticated (though not well explained), and the estimates are remarkably close to those from previous studies, and estimated by less direct methods.

Overall, this is a large-scale study that addresses an important issue: whether quantitative variation is maintained by a balance between mutation and selection. The results are not surprising, and support the consensus view, but that should not count against this work. On balance, I think that it should be suitable for eLife, but it does need some considerable revision – mostly, to explain what was done more clearly to a reader who has not been immersed in these issues.

We have extensively revised the manuscript, thanks to constructive comments from the editor and reviewers. One main revision was to explain better the experimental design and expectation of the experiments, as this reviewer suggests.

Generally, the paper is written for readers thoroughly familiar with these kinds of experiment. There is some explanation in the Discussion, but it needs to come much earlier. There needs to be a clearer statement early on about what is expected in mutation accumulation experiments. Highly deleterious mutations will not be seen, but weakly deleterious mutations will accumulate in the MA lines, and will be more frequent than in nature.

The estimates here do not include the fraction of strongly deleterious mutations. In particular, one expects recessive deleterious mutations to potentially rise to moderate frequency in these inbred lines. This would not necessarily show up in Figure 1 if the bulk of mutations are nearly neutral, but could be tested for by asking whether the MA lines can be made homozygous. A comment at least is needed on this issue.

This is a known limitation of MA lines, which must be free of lethal and highly deleterious mutations to keep them alive. Furthermore, beneficial mutations tend to fix at a higher probability thus biasing mutation rate estimates upwards in MA lines. These limitations have now been acknowledged both in the Introduction and in Discussion

A pervasive problem is the use of the term "stabilising selection" to refer to any kind of purifying selection. It would be clearer to restrict it to selection on a continuous trait towards an intermediate optimum, and it needs t be stead always whether this is selection on the observed or on hidden traits. In any case, it is simpler to see V_m_/V_g_ as an estimate of net selection against the alleles that contribute trait variance, and avoid speculation about whether this acts via stabilising selection (in the sense defined here).

We have revised the text extensively to refer to the stabilizing selection indicated by Vm/Vg as apparent stabilizing selection and distinguish between stabilizing selection due to direct fitness effects for a trait or pleiotropic effects where possible.

Results, first paragraph: It would be interesting to compare the frequency of accumulated mutations with the spontaneous mutation rate, measured in other studies that compare parents with offspring – the difference being that the MA lines will not contain highly deleterious mutations.

This comparison would certainly be very interesting. However, it is not as simple as it appears and goes beyond the scope of this study. First, there has been only a limited number of de novo mutations detected in parent-offspring trio data in *Drosophila* (e.g. Keightley, et al., Genetics, 2014). The mutation rate estimated using a full-sib family agreed within the 95% confidence intervals with ours (Keightley, et al. 2014), suggesting that the mutation rate was not drastically higher in inbred lines. However, there were only eight mutations in total detected and confirmed in the Keightley (2014) study, which brings us to the second point. Even if there were such trio data, because of the small mutation rate, the number of mutations present in a single fly genome is small (on average ~1-2 mutations per fly). This requires an enormous number of flies to be sequenced to enable a meaningful comparison. This is precisely the point of doing mutation accumulation, which is to accumulate mutations in the MA lines so there characteristics can be properly studied.

A serious worry is whether mutation rate is higher, and different in kind, in inbred lines than in the wild population – cf Agrawal's recent work.

This is a problem that goes beyond the scope of the present study. We now note this limitation and its implications in the Discussion. The bottom line is, it is unlikely that this mutation rate bias in inbred lines alone is able to explain the large magnitude of strong apparent stabilizing selection.

Results,second paragraph: "Generally agreed well" is rather vague – one needs a statistical test of whether the rate and spectrum of mutations is the same as in previous work (as SI).

We changed the statement to include specific numbers from previous work. Regarding the test of the spectrum of mutations, we do not expect exact agreement between this work and earlier studies because different bioinformatic procedures can have significant effects on the inference of the mutation spectrum. Nevertheless, we detected 162 A/T->G/C, 91 A/T->C/G, 148 A/T->T/A, 433 G/C->A/T, 158 G/C->C/G, 211 G/C->T/A mutations while Keightley et al. (2009) detected 33, 20, 21, 53, 20, 27 (Table 5 of the reference) such mutations respectively, which had a Fisher’s exact test P value of 0.16. Thus our mutations and theirs do not have characteristic difference. However, we choose not to include this in the manuscript to avoid false expectation.

Subsection “Spontaneous mutation rate”: Given the population size, what is the fixation probability as a function of h and s? As it is, the reader has to work this out for herself.

We added to the manuscript a range of fixation probabilities given a range of. For simplicity, we assumed additivity so that a one dimensional range can be provided. It is important to note that, because of the small population size, for small to moderate s, the fixation probability is very close to the neutral probability, which is 1/2N.

In the same section, there should be a bit more explanation of how the mutation rate is estimated in the main text (a sentence, at least). This is not at all straightforward.

According to the reviewer’s suggestion, we added a brief description of the mutation rate estimation process in the main text.

Subsection “Rate of introduction of genetic variation by spontaneous mutations”: What is meant by "global gene expression"? Even after reading the Methods, I could not work this out. It seems bizarre to pool the expression level of all the genes, because, I assume, expression of individual genes will be analysed elsewhere. Still, it is not clear what "global gene expression" is measuring, especially since the expression measurements were made separately in the three experiments and so are not directly comparable.

We apologize for the confusing term. We mean genome-wide gene expression profiles. The “global” was meant to highlight the genome-wide characteristic of microarrays. To avoid confusion, we replace this term with “genome-wide gene expression profiles”.

Second paragraph of the same section: What are the units of measurement here? It is not obvious how to compare variance in gene expression, but the definition of the trait is not even stated in the main text. In the Methods, it emerges that there has been a very obscure normalisation procedure. Are we comparing the variance of absolute values? The variance on a log scale? The variance relative to V_e_? This is crucial to how we interpret these variances.

The measurement unit is the intensity of hybridization followed by logarithm transformation to make the distribution more or less normal. This has been the standard practice in obtaining expression levels from microarrays.

Regarding the questions the reviewer has:

1) The variance is not on absolute values; 2) the expression level is on a log scale, the variance is the variance of the expression level; 3) we scale Vm by Ve to obtain mutational heritability hm2. All other comparisons are directly on and can only be on Vmor Vgbecause the two experiments have different Ve.

Subsection “Strong stabilizing selection on quantitative trait variation”: Seeing lower standing variance than expected under neutrality does not imply stabilising selection; there could instead be purifying selection against alleles that have pleiotropic effects on the measured traits. The term "stabilising selection" needs to be defined carefully.

We appreciate this comment. To clarify things, we have now make explicit distinctions between real stabilizing selection and apparent stabilizing selection that may be the result of pleiotropic effects.

*In the same section, the comparison of*
E(am2)
*vs*
E(ag2)
*is confusing. The implicit assumption here is that stabilising selection acts on the observed traits, which is wildly implausible. Rather, V_m_/V_g_ gives an estimate of the net selection on each allele. This selection may be due to stabilising selection on unobserved traits, and/or to direct selection against the alleles. (The distinction here depends of course on what "traits" one defines). It is confusing to use the term "stabilising selection" here.*

We now refer to the observed variance difference as apparent stabilizing selection. Nevertheless, we still find the comparison of E(ag2) and E(am2) useful because they provide a connection between the DNA sequences and expression profiles in this study, which has not been possible before. It is perhaps best to consider these allelic effects as the “apparent” or “net” (as the reviewer used) effects, which for pleiotropic effects, are simply the indirect effects of the fitness changing DNA variants on quantitative traits. Regardless of the nature of the stabilizing selection, the outcome is that the effects (whether they are direct or indirect) they have on the traits are on average smaller for standing variation than spontaneous mutations, which we think is an important result. To avoid confusion, we have revised the relevant text.

The basic theory is explained in the Discussion but needs to come much earlier to avoid confusion.

We have revised the text extensively to provide better introduction to the experimental design (e.g. Figure 1) and explanation of results.

Discussion, paragraph four: The estimate of selection from V_m_/V_g_ is an order of magnitude smaller than estimates of selection against deleterious mutations generally. However, this is consistent with a broad distribution of effects on fitness and on traits: the alleles that contribute to genetic variance will tend to be less deleterious than spontaneous mutations. Also, of course, balancing selection may contribute genetic variance.

We agree with the reviewer. These points were in fact made in the text that came a paragraph later, where we offered alternative explanations.

Subsection “Estimation of mutation rate in MA lines”: Does the actual frequency distribution of (presumed) neutral sites actually fit the simulation model? Figure 6 suggests yes, but this is not commented on in the text.

We have added this description to the text.

It seems rather inefficient to just record the number of mutations at p>0.2 – surely using the actual frequency would be more informative? One would also like to see whether non-synonymous mutations are less likely to reach high frequency, as expected if they are recessive deleterious.

We wish we could reliably detect mutations of lower frequency. The sequencing was done on a pool of DNA and was not without error. Choosing a higher frequency cut-off allows us to more accurately identify mutations and estimate mutation rates.

Subsection “Variance in quantitative traits due to mutations, standing variation, and environmental variation”: This formula for VMA makes no sense; it will be negative for large times. There is no explanation of where it comes from.

We are not entirely sure why this formula would lead to negative values for large times. For many numerical examples we tried, k is closest to zero for large Ne and small t, in which the second term in the bracket is a very small number, making k positive. This formula is taken from the reference Mackay et al. 1992, which was based on formula in Lynch and Hill, 1986. These two papers were cited at the end of the sentence introducing this formula.

It is annoying that the figures are all given at the end, but are separated from their captions, and are not numbered.

We agree. This has to do with the journal’s submission system, which asks for separate files for figures. The PDF file is then generated automatically by concatenating figures to the end of the text.

Figure 7 and Figure 9 are especially inscrutable.

We apologize for the crowded text and bars. We have enlarged the figures to better show the bars and texts. We hope the figures are more readable now.

[Editors' note: further revisions were requested prior to acceptance, as described below.]

Reviewer #3:

The revision more or less answers my criticisms, though the changes are minimal. I accept that starting from a single line was a reasonable choice, given the effort involved, and this is now explained better. However, I am still not satisfied on a couple of points:

The revisions have been mostly to better explain the experimental design. However, my original criticism was more that the underlying theory is not set out well at the beginning: this has not been addressed.

A specific problem is with a key interpretation of the results: in paragraph seven of the Discussion it's stated that the estimated selection is too weak to be consistent with classical estimates of selection against alleles that influence viability. I don't think this necessarily follows. If there is a distribution of effects on traits, then the ratio V_m_/V_g_ will be smaller for traits that are less closely connected with fitness. With rare alleles, and assuming mutation/selection balance, the selection estimated from V_m_/V_g_ equals the harmonic mean s, weighted by the squared affect on traits. Thus, of the effect on the trait is not correlated with fitness, we have 1/<1/s>, whereas if the trait is fitness itself, we have <s^2>/<s>, which must be larger. So, the differences in V_m_/V_g_ between viability and gene expression are surely to be expected?

These issues are discussed in Charlesworth's recent PNAS review, which should be not just cited, but used to frame the interpretation of these data.

A point of terminology: I don't think that "apparent stabilising selection" is used correctly. To me, this refers to the reduction in fitness of extreme phenotypes that is observed even for a neutral trait, when trait variation is due to pleiotropic effects of mutations. However, here "apparent SS" refers to the ratio V_m_/V_g_ under this pleiotropic model. I think that "purifying" or "pleiotropic" selection would be a better term to use.

We thank this reviewer for the comments on the interpretation and presentation of data. We respectively disagree with some of these points. Specifically:

1) Regarding how to frame the interpretation of the data, we frame this work as more of an empirical work rather than theoretical. To this end and following the reviewers’ comments in the previous revision, we have provided detailed information on the rationale and expectation of the experimental outcomes. We believe this is of sufficiently clarity. While we are aware of the discrepancies between the data and theories and the fact that there have been presently no consensus theory to reconcile the two, we discuss the implications at the end of the manuscript in length, hoping that the ongoing debate won’t distract the main experimental results.

2) For the difference between *Vm/Vg* for gene expression and fitness, we agree with the reviewer to the extent that some difference is expected because not all gene expression traits are directly correlated with fitness. There is a distribution of effects and by using the median as the numerical example, we are making the point that at least 50% of the genes are experiencing selection that’s weaker than selection on fitness. We are not entirely sure why this causes an issue.

3) Finally, we believe the term “apparent stabilizing selection” faithfully reflects the nature of the data, i.e., the selection apparently constrains variance but the form is unknown. Purifying or pleiotropic selection are too specific terms to be supported by our data.

4) Charlesworth’s excellent review was specifically for effects of mutations on fitness and its components, not the quantitative traits examined in this manuscript. The point that the strength of selection implicated by the empirical estimate of *Vm/Vg* is too small by an order of magnitude to be consistent with mutation-apparent (pleiotropic) stabilizing selection has been made before (e.g. Barton 1990) and is generally accepted, so we do not understand the reviewer’s objection to this interpretation.